# GnnX-Bench: Unravelling the Utility of Perturbation-based Gnn Explainers through In-depth Benchmarking

**Mert Kosan**[1][*][†], **Samidha Verma**[2][*], **Burouj Armgaan**[2], **Khushbu Pahwa**[3], **Ambuj Singh**[1]
**Sourav Medya**[4], **Sayan Ranu**[2]
University of California, Santa Barbara[1]
Indian Institute of Technology, Delhi[2]
Rice University[3]
University of Illinois, Chicago[4]
mertkosan@gmail.com, kp66@rice.edu, ambuj@cs.ucsb.edu, medya@uic.edu
{samidha.verma, burouj.armgaan, sayanranu}@cse.iitd.ac.in

## Abstract

Numerous explainability methods have been proposed to shed light on the inner workings of Gnns. Despite the inclusion of empirical evaluations in all the proposed algorithms, the interrogative aspects of these evaluations lack diversity. As a result, various facets of explainability pertaining to Gnns, such as a comparative analysis of counterfactual reasoners, their stability to variational factors such as different Gnn architectures, noise, stochasticity in non-convex loss surfaces, feasibility amidst domain constraints, and so forth, have yet to be formally investigated. Motivated by this need, we present a benchmarking study on perturbation-based explainability methods for Gnns, aiming to systematically evaluate and compare a wide range of explainability techniques. Among the key findings of our study, we identify the Pareto-optimal methods that exhibit superior efficacy and stability in the presence of noise. Nonetheless, our study reveals that all algorithms are affected by stability issues when faced with noisy data. Furthermore, we have established that the current generation of counterfactual explainers often fails to provide feasible recourses due to violations of topological constraints encoded by domain-specific considerations. Overall, this benchmarking study empowers stakeholders in the field of Gnns with a comprehensive understanding of the state-of-the-art explainability methods, potential research problems for further enhancement, and the implications of their application in real-world scenarios.

## 1 Introduction and Related Work

Gnns have shown state-of-the-art performance in various domains including social networks Manchanda et al. (2020); Chakraborty et al. (2023), biological sciences Ying et al. (2021); Rampášek et al. (2022); Ranjan et al. (2022), modeling of physical systems Thangamuthu et al. (2022); Bhattoo et al. (2022; 2023); Bishnoi et al. (2023), event detection Cao et al. (2021); Kosan et al. (2021) and traffic modeling Gupta et al. (2023); Jain et al. (2021); Wu et al. (2017); Li et al. (2020). Unfortunately, like other deep-learning models, Gnns are black boxes due to lacking transparency and interpretability. This lack of interpretability is a significant barrier to their adoption in critical domains such as healthcare, finance, and law enforcement. In addition, the ability to explain predictions is critical towards understanding potential flaws in the model and generate insights for further refinement. To impart interpretability to Gnns, several algorithms to explain the inner workings of Gnns have been proposed. The diversified landscape of Gnn explainability research is visualized in Fig. 1. We summarize each of the categories below:

- **Model-level:** Model-level or global explanations are concerned with the overall behavior of the model and search for patterns in the set of predictions made by the model. XGNN Yuan et al. (2020), GLG-Explainer Azzolin et al. (2023), Xuanyuan et al. Xuanyuan et al. (2023), GCFExplainer Huang et al. (2023).

---

*Both authors contributed equally to this research.
†Work done prior to joining Visa Inc.

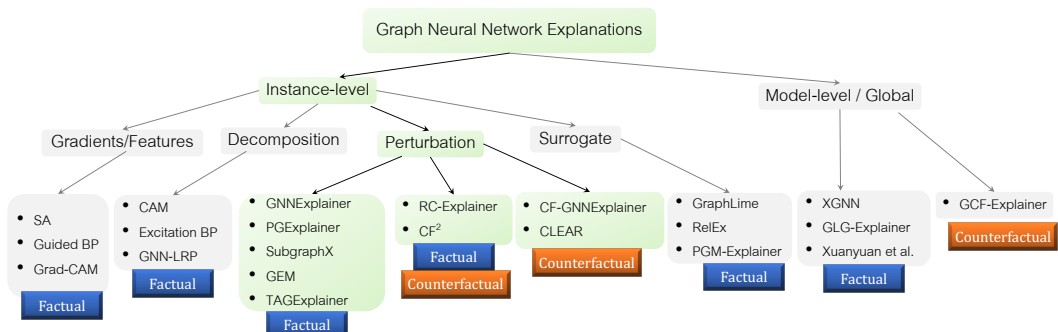

Figure 1: Structuring the space of the existing methods on GNN explainability.

- **Instance-level:** Instance-level or local explainers provide explanations for specific predictions made by a model. For instance, these explanations reason why a particular instance or input is classified or predicted in a certain way.
- **Gradient-based:** They follow the idea of the rate of change being represented by gradients. Additionally, the gradient of the prediction with respect to the input represents the prediction sensitivity to the input. This sensitivity gives the importance scores and helps in finding explanations. SA and Guided-BP Baldassarre & Azizpour (2019), Grad-CAM Pope et al. (2019).
- **Decomposition-based:** They consider the prediction of the model to be decomposed and distributed backward in a layer-by-layer fashion and the score of different parts of the input can be construed as its importance to the prediction. CAM and Excitation-BP Pope et al. (2019), GNN-LRP Schnake et al. (2021).
- **Perturbation-based:** They utilize input perturbations to identify important subgraphs serving as factual or counterfactual explanations. GNNExplainer Ying et al. (2019b), PGExplainer Luo et al. (2020), SubgraphX Yuan et al. (2021), GEM Lin et al. (2021a), TAGExplainer Xie et al. (2022), $CF^2$ Tan et al. (2022), RCExplainer Bajaj et al. (2021), CF-GNNexplainer Lucic et al. (2022), CLEAR Ma et al. (2022), Shan et al. (2021); Abrate & Bonchi (2021); Wellawatte et al. (2022)
- **Surrogate:** They use the generic intuition that in a smaller range of input values, the relationship between input and output can be approximated by interpretable functions. The methods fit a simple and interpretable surrogate model in the locality of the prediction. GraphLime Huang et al. (2022), Relex Zhang et al. (2021), PGM-Explainer Vu & Thai (2020).

The type of explanation offered represents a crucial component. Explanations can be broadly classified into two categories: *factual* reasoning and *counterfactual* reasoning.

- **Factual explanations** provide insights into the rationale behind a specific prediction by identifying the minimal subgraph that is sufficient to yield the same prediction as the entire input graph.
- **Counterfactual explanations** elucidate why a particular prediction was not made by presenting alternative scenarios that could have resulted in a different decision. In the context of graphs, this involves identifying the smallest perturbation to the input graph that alters the prediction of the GNN. Perturbations typically involve the removal of edges or modifications to node features.

## 1.1 CONTRIBUTIONS

In this benchmarking study, we systematically study perturbation-based factual and counterfactual explainers and identify their strengths and limitations in terms of their ability to provide accurate, meaningful, and actionable explanations for GNN predictions. The proposed study surfaces new insights that have not been studied in existing benchmarking literature Amara et al. (2022); Agarwal et al. (2023)(See. App. J for details). Overall, we make the following key contributions:

- **Comprehensive evaluation encompassing counterfactual explainers:** The benchmarking study encompasses seven factual explainers and four counterfactual explainers. The proposed work is the first benchmarking study on counterfactual explainers for GNNs.
- **Novel insights:** The findings of our benchmarking study unveil stability to noise and variational factors and generating feasible counterfactual recourses as two critical technical deficiencies that naturally lead us towards open research challenges.
- **Codebase:** As a by-product, a meticulously curated, publicly accessible code base is provided (https://github.com/idea-iitd/gnn-x-bench/).

Table 1: Key highlights of the *perturbation-based* factual methods. The "NFE" column implies *Node Feature Explanation*. "GC" and "NC" indicate whether the dataset is used for graph classification and node classification respectively.

| Method | Subgraph Extraction Strategy | Scoring function | Constraints | NFE | Task | Nature |
|---|---|---|---|---|---|---|
| GNNExplainer | Continuous relaxation | Mutual Information | Size | Yes | GC+NC | Transductive |
| PGExplainer | Parameterized edge selection | Mutual Information | Size, Connectivity | No | GC+NC | Inductive |
| TAGExplainer | Sampling | Mutual Information | Size, Entropy | No | GC+NC | Inductive |
| GEM | Granger Causality+Autoencoder | Causal Contribution | Size, Connectivity | No | GC+NC | Inductive |
| SubgraphX | Monte Carlo Tree Search | Shapley Value | Size, Connectivity | No | GC | Transductive |

## 2 PRELIMINARIES AND BACKGROUND

We use the notation $\mathcal{G} = (\mathcal{V}, \mathcal{E})$ to represent a graph, where $\mathcal{V}$ denotes the set of nodes and $\mathcal{E}$ denotes the set of edges. Each node $v_i \in \mathcal{V}$ is associated with a feature vector $x_i \in \mathbb{R}^d$. We assume there exists a GNN $\Phi$ that has been trained on $\mathcal{G}$ (or a set of graphs).

The literature on GNN explainability has primarily focused on *graph classification* and *node classification*, and hence the output space is assumed to be categorical. In graph classification, we are given a set of graphs as input, each associated with a class label. The task of the GNN $\Phi$ is to correctly predict this label. In the case of node classification, class labels are associated with each node and the predictions are performed on nodes. In a message passing GNN of $\ell$ layers, the embedding on a node is a function of its $\ell$-hop neighborhood. We use the term *inference subgraph* to refer to this $\ell$-hop neighborhood. Henceforth, we will assume that graph refers to the inference subgraph for node classification. Factual and counterfactual reasoning over GNNs are defined as follows.

**Definition 1 (Perturbation-based Factual Reasoning)** *Let $\mathcal{G}$ be the input graph and $\Phi(\mathcal{G})$ the prediction on $\mathcal{G}$. Our task is to identify the smallest subgraph $\mathcal{G}_S \subseteq \mathcal{G}$ such that $\Phi(\mathcal{G}) = \Phi(\mathcal{G}_S)$. Formally, the optimization problem is expressed as follows:*

$$\mathcal{G}_S = \arg \min_{\mathcal{G}' \subseteq \mathcal{G}, \, \Phi(\mathcal{G}) = \Phi(\mathcal{G}')} ||\mathcal{A}(\mathcal{G}')|| \qquad (1)$$

*Here, $\mathcal{A}(\mathcal{G}_S)$ denotes the adjacency matrix of $\mathcal{G}_S$, and $||\mathcal{A}(\mathcal{G}_S)||$ is its L1 norm which is equivalent to the number of edges. Note that if the graph is undirected, the number of edges is half of the L1 norm. Nonetheless, the optimization problem remains the same.*

While subgraph generally concerns only the topology of the graph, since graphs in our case may be annotated with features, some algorithms formulate the minimization problem in the joint space of topology and features. Specifically, in addition to identifying the smallest subgraph, we also want to minimize the number of features required to characterize the nodes in this subgraph.

**Definition 2 (Counterfactual Reasoning)** *Let $\mathcal{G}$ be the input graph and $\Phi(\mathcal{G})$ the prediction on $\mathcal{G}$. Our task is to introduce the minimal set of perturbations to form a new graph $\mathcal{G}^*$ such that $\Phi(\mathcal{G}) \neq \Phi(\mathcal{G}^*)$. Mathematically, this entails to solving the following optimization problem.*

$$\mathcal{G}^* = \arg \min_{\mathcal{G}' \in \mathbb{G}, \, \Phi(\mathcal{G}) \neq \Phi(\mathcal{G}')} dist(\mathcal{G}, \mathcal{G}') \qquad (2)$$

*where $dist(\mathcal{G}, \mathcal{G}')$ quantifies the distance between graphs $\mathcal{G}$ and $\mathcal{G}'$ and $\mathbb{G}$ is the set of all graphs one may construct by perturbing $\mathcal{G}$. Typically, distance is measured as the number of edge perturbations while keeping the node set fixed. In case of multi-class classification, if one wishes to switch to a target class label(s), then the optimization objective is modified as $\mathcal{G}^* = \arg \min_{\mathcal{G}' \in \mathbb{G}, \, \Phi(\mathcal{G}') = \mathbb{C}} dist(\mathcal{G}, \mathcal{G}')$, where $\mathbb{C}$ is the set of desired class labels.*

### 2.1 REVIEW OF PERTURBATION-BASED GNN REASONING

**Factual (Yuan et al. (2022); Kakkad et al. (2023)):** The perturbation schema for factual reasoning usually consists of two crucial components: the subgraph extraction module and the scoring function module. Given an input graph $\mathcal{G}$, the subgraph extraction module extracts a subgraph $\mathcal{G}_s$; and the scoring function module evaluates the model predictions $\Phi(\mathcal{G}_s)$ for the subgraphs, comparing them with the actual predictions $\Phi(\mathcal{G})$. For instance, while GNNExplainer Ying et al. (2019a) identifies an explanation in the form of a subgraph that have the maximum influence on the prediction, PG-Explainer Luo et al. (2020) assumes the graph to be a random Gilbert graph. Unlike the existing explainers, TAGExplainer Xie et al. (2022) takes a two-step approach where the first step has an

Table 2: Key highlights of the counterfactuals methods."GC" and "NC" indicate whether the dataset is used for graph classification and node classification respectively.

| Method | Explanation Type | Task | Target/Method | Nature |
|---|---|---|---|---|
| RCExplainer Bajaj et al. (2021) | Instance level | GC+NC | Neural Network | Inductive |
| CF$^2$ Tan et al. (2022) | Instance level | GC+NC | Original graph | Transductive |
| CF-GNNExplainer Lucic et al. (2022) | Instance level | NC | Inference subgraph | Transductive |
| CLEAR Ma et al. (2022) | Instance level | GC+NC | Variational Autoencoder | Inductive |

embedding explainer trained using a self-supervised training framework without any information of the downstream task. On the other hand, GEM Lin et al. (2021a) uses Granger causality and an autoencoder for the subgraph extraction strategy where as SubgraphX Yuan et al. (2021) employes a monte carlo tree search. The scoring function module uses mutual information for GNNExplainer, PGExplainer, and TAGExplainer. This module is different for GEM and SubgraphX, and uses casual contribution and Shapley value respectively. Table 1 summarizes the key highlights.

**Counterfactual (Yuan et al. (2022)):** The four major counterfactual methods are CF-GNNExplainer Lucic et al. (2022), CF$^2$ Tan et al. (2022), RCExplainer Bajaj et al. (2021), and CLEAR Ma et al. (2022). They are instance-level explainers and apply to both graph and node classification tasks except for CF-GNNExplainer which is only applied to node classification. In terms of method, CF-GNNExplainer aims to perturb the computational graph by using a binary mask matrix. The corresponding loss function quantifies the accuracy of the produced counterfactual and captures the distance (or similarity) between the counterfactual graph and the original graph, whereas, CF$^2$ Tan et al. (2022) extends this method by including a contrastive loss that jointly optimizes the quality of both the factual and the counterfactual explanation. Both of the above methods are transductive. As an inductive method, RCExplainer Bajaj et al. (2021), aims to identify a resilient subset of edges to remove such that it alters the prediction of the remaining graph while CLEAR Ma et al. (2022) generates counterfactual graphs by using a graph variational autoencoder. Table 2 summarizes the key highlights.

## 3 BENCHMARKING FRAMEWORK

In this section, we outline the investigations we aim to conduct and the rationale behind them. The mathematical formulation of the various metrics are summarized in Table 3.

**Comparative Analysis:** We evaluate algorithms for both factual and counterfactual reasoning and identify the pareto-optimal methods. The performance is quantified using *explanation size* and *sufficiency* Tan et al. (2022). Sufficiency encodes the ratio of graphs for which the prediction derived from the explanation matches the prediction obtained from the complete graph Tan et al. (2022). Its value spans between 0 and 1. For factual explanations, higher values indicate superior performance, while in counterfactual lower is better since the objective is to flip the class label.

**Stability:** Stability of explanations, when faced with minor variations in the evaluation framework, is a crucial aspect that ensures their reliability and trustworthiness. Stability is quantified by taking the *Jaccard similarity* between the set of edges in the original explanation vs. those obtained after introducing the variation (details in § 4). In order to evaluate this aspect, we consider the following perspectives:

- **Perturbations in topological space:** If we inject minor perturbations to the topology through a small number of edge deletions or additions, then that should not affect the explanations.
- **Model parameters:** The explainers are deep-learning models themselves and optimize a non-convex loss function. As a consequence of non-convexity, when two separate instances of the explainer starting from different seeds are applied to the same GNN model, they generate dissimilar

Table 3: The various metrics used to benchmark the performance of GNN explainers.

| | |
|---|---|
| Sufficiency$(\mathcal{S}) = \frac{\sum_{i=1}^{\|\mathbb{G}\|} \mathbb{1}(\Phi(\mathcal{G}_S^i)=\Phi(\mathcal{G}^i))}{\|\mathbb{G}\|}$ | • $\mathbb{G} = \{\mathcal{G}^1, \mathcal{G}^2, \ldots, \mathcal{G}^n\}$: graph set. |
| | • $\mathcal{G}_S^i$: explanation subgraph of $\mathcal{G}^i$ |
| Necessity$(\mathcal{N}) = \frac{\sum_{i=1}^{\|\mathbb{G}\|} \mathbb{1}(\Phi(\mathcal{R}^i)\neq\Phi(\mathcal{G}^i))}{\|\mathbb{G}\|}$ | • $\mathbb{G}_S = \{\mathcal{G}_S^1, \mathcal{G}_S^2, \ldots, \mathcal{G}_S^n\}$: explanation set. |
| | • $\mathcal{R}^i = \mathcal{G} - \mathcal{G}_S^i$ |
| Stability$(\mathcal{E}_X, \mathcal{E}_X') = \frac{\|\mathcal{E}_X \cap \mathcal{E}_X'\|}{\|\mathcal{E}_X \cup \mathcal{E}_X'\|}$ | • $\mathbb{R} = \{\mathcal{R}^1, \mathcal{R}^2, \ldots, \mathcal{R}^n\}$: residual graph set. |
| | • $\Phi, \Phi_S, \Phi_R$: the models trained on $\mathbb{G}, \mathbb{G}_S, \mathbb{R}$. |
| Reproducibility$^+(\mathcal{R}^+) = \frac{ACC(\Phi_S)}{ACC(\Phi)}$ | All models are trained on the same labels. |
| | • $\Phi(\mathbb{G}^i)$: the prediction of the model on $\mathbb{G}^i$. |
| Reproducibility$^-(\mathcal{R}^-) = \frac{ACC(\Phi_R)}{ACC(\Phi)}$ | • $ACC(\Phi)$: the test accuracy of $\Phi$. |

explanations. Our benchmarking study investigates the impact of this stochasticity on the quality and consistency of the explanations produced.

- **Model architectures:** Message-passing GNNs follow a similar computation framework, differing mainly in their message aggregation functions. We explore the stability of explanations under variations in the model architecture.

**Necessity:** Factual explanations are *necessary* if the removal of the explanation subgraph from the graph results in counterfactual graph (i.e., flipping the label).

**Reproducibility:** We measure two different aspects related to how central the explanation is towards retaining the prediction outcomes. Specifically, Reproducibility$^+$ measures if the GNN is retrained on the explanation graphs alone, can it still obtain the original predictions? On the other hand, Reproducibility$^-$ measures if the GNN is retrained on the *residual* graph constructed by removing the explanation from the original graph, can it still predict the class label? The mathematical quantification of these metrics is presented in Fig. 3.

**Feasibility:** One notable characteristic of counterfactual reasoning is its ability to offer recourse options. Nonetheless, in order for these recourses to be effective, they must adhere to the specific domain constraints. For instance, in the context of molecular datasets, the explanation provided must correspond to a valid molecule. Likewise, if the domain involves consistently connected graphs, the recourse must maintain this property. The existing body of literature on counterfactual reasoning with GNNs has not adequately addressed this aspect, a gap we address in our benchmarking study.

Table 4: The statistics of the datasets. Here, "F" and "CF" in the column "'X-type" indicates whether the dataset is used for Factual or Counterfactual reasoning. "GC" and "NC" in the *Task* column indicates whether the dataset is used for graph classification and node classification respectively.

| | #Graphs | #Nodes | #Edges | #Features | #Classes | Task | F/CF |
|---|---|---|---|---|---|---|---|
| MUTAGENICITY Riesen & Bunke (2008); Kazius et al. (2005) | 4337 | 131488 | 133447 | 14 | 2 | GC | F+CF |
| PROTEINS Borgwardt et al. (2005); Dobson & Doig (2003) | 1113 | 43471 | 81044 | 32 | 2 | GC | F+CF |
| IMDB-B Yanardag & Vishwanathan (2015) | 1000 | 19773 | 96531 | 136 | 2 | GC | F+CF |
| AIDS Ivanov et al. (2019) | 2000 | 31385 | 32390 | 42 | 2 | GC | F+CF |
| MUTAG Ivanov et al. (2019) | 188 | 3371 | 3721 | 7 | 2 | GC | F+CF |
| NCI1 Wale et al. (2008) | 4110 | 122747 | 132753 | 37 | 2 | GC | F |
| GRAPH-SST2 Yuan et al. (2022) | 70042 | 714325 | 644283 | 768 | 2 | GC | F |
| DD Dobson & Doig (2003) | 1178 | 334925 | 843046 | 89 | 2 | GC | F |
| REDDIT-B Yanardag & Vishwanathan (2015) | 2000 | 859254 | 995508 | 3063 | 2 | GC | F |
| OGBG-MOLHIV Allamanis et al. (2018) | 41127 | 1049163 | 2259376 | 9 | 2 | GC | CF |
| TREE-CYCLES Ying et al. (2019a) | 1 | 871 | 1950 | 10 | 2 | NC | CF |
| TREE-GRID Ying et al. (2019a) | 1 | 1231 | 3410 | 10 | 2 | NC | CF |
| BA-SHAPES Ying et al. (2019a) | 1 | 700 | 4100 | 10 | 4 | NC | CF |

## 4 EMPIRICAL EVALUATION

In this section, we execute the investigation plan outlined in § 3. Unless mentioned specifically, the base black-box GNN is a GCN. Details of the set up (e.g., hardware) are provided in App. A.

**Datasets:** Table 4 showcases the principal statistical characteristics of each dataset employed in our experiments, along with the corresponding tasks evaluated on them. The TREE-CYCLES, TREE-GRID, and BA-SHAPES datasets serve as benchmark graph datasets for counterfactual analysis. These datasets incorporate ground-truth explanations Tan et al. (2022); Lin et al. (2021a); Lucic et al. (2022). Each dataset contains an undirected base graph to which predefined motifs are attached to random nodes, and additional edges are randomly added to the overall graph. The class label assigned to a node determines its membership in a motif.

### 4.1 COMPARATIVE ANALYSIS

**Factual Explainers:** Fig. 2 illustrates the sufficiency analysis of various factual reasoners in relation to size. Each algorithm assigns a score to edges, indicating their likelihood of being included in the factual explanation. To control the size, we adopt a greedy approach by selecting the highest-scoring edges. Both CF$^2$ and RCEXPLAINER necessitate a parameter to balance factual and counterfactual explanations. We set this parameter to 1, corresponding to solely factual explanations.

**Insights:** No single technique dominates across all datasets. For instance, while RCEXPLAINER performs exceptionally well in the MUTAG dataset, it exhibits subpar performance in IMDB-B and GRAPH-SST2. Similar observations are also made for GNNEXPLAINER in REDDIT-B vs. MUTAG and NCI1. Overall, we recommend using either RCEXPLAINER or GNNEXPLAINER as the preferred choices. The spider plot in Fig. Q more prominently substantiates this suggestion. GNNEXPLAINER is transductive, wherein it trains the parameters on the input graph itself. In contrast, inductive

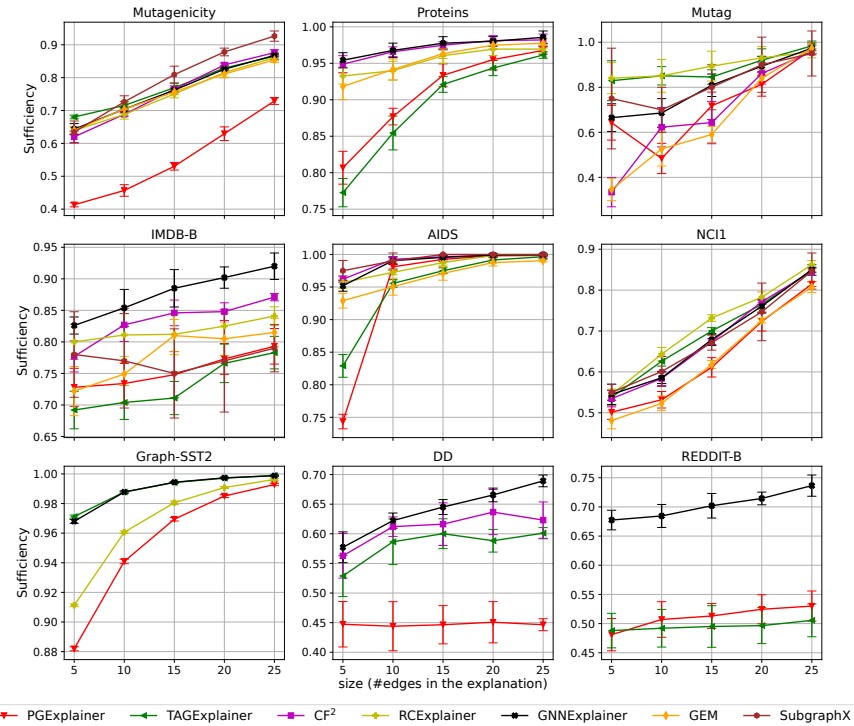

Figure 2: Sufficiency of the factual explainers against the explanation size. For factual explanations, higher is better. We omit those methods for a dataset that threw an out-of-memory (OOM) error.

methods use pre-trained weights to explain the input. Consequently, transductive methods, such as GNNEXPLAINER, at the expense of higher computation cost, has an inherent advantage in terms of optimizing sufficiency. Compared to other transductive methods, GNNEXPLAINER utilizes a loss function that aims to increase sufficiency directly. This makes the method a better candidate for sufficiency compared to other inductive and transductive explainers. On the other hand, for RCEXPLAINER, we believe that calculation of decision regions for classes helps to increase its generalizability as well as robustness.

In Fig. 2, the sufficiency does not always increase monotonically with explanation size (such as PGEXPLAINER in Mutag). This behavior arises due to the combinatorial nature of the problem. Specifically, the impact of adding an edge to an existing explanation on the GNN prediction is a function of both the edge being added and the edges already included in the explanation. An explainer seeks to learn a proxy function that mimics the true combinatorial output of a set of edges. When this proxy function fails to predict the marginal impact of adding an edge, it could potentially select an edge that exerts a detrimental influence on the explanation's quality.

Table 5: Sufficiency and size of counterfactual explainers on graph classification. Lower values are better for both metrics. OOM indicates that the technique ran out of memory.

| Method / Metric | Mutag | | Mutagenicity | | AIDS | | Proteins | | IMDB-B | | ogbg-molhiv | |
| | Suffic.↓ | Size↓ | Suffic.↓ | Size↓ | Suffic.↓ | Size↓ | Suffic.↓ | Size↓ | Suffic.↓ | Size↓ | Suffic.↓ | Size↓ |
|---|---|---|---|---|---|---|---|---|---|---|---|---|
| **RCExplainer** | 0.4 ± 0.12 | 1.1 ± 0.22 | 0.4 ± 0.06 | 1.01 ± 0.19 | 0.91 ± 0.04 | 1.0 ± 0.0 | 0.96 ± 0.02 | 1.0 ± 0.0 | 0.72 ± 0.11 | 1.0 ± 0.0 | 0.90 ± 0.02 | 1 ± 0.0 |
| **CF²**($\alpha = 0$) | 0.90 ± 0.12 | 1.0 ± 0.0 | 0.50 ± 0.05 | 2.78 ± 0.98 | 0.98 ± 0.02 | 5.25 ± 0.35 | 1.0 ± 0.0 | NA | 0.81 ± 0.07 | 8.57 ± 4.99 | 0.96 ± 0.00 | 10.45 ± 4.43 |
| **CLEAR** | 0.55 ± 0.1 | 17.15 ± 1.62 | OOM | OOM | 0.84 ± 0.03 | 164.9 ± 47.9 | OOM | OOM | 0.96 ± 0.02 | 218.62 ± 0 | OOM | OOM |

**Counterfactual Explainers:** Tables 5 and 6 present the results on graph and node classification.

**Insights on graph classification (Table 5):** RCEXPLAINER is the best-performing explainer across the majority of the datasets and metrics. However, it is important to acknowledge that RCEXPLAINER's sufficiency, when objectively evaluated, consistently remains high, which is undesired. For instance, in the case of AIDS, the sufficiency of RCEXPLAINER reaches a value of 0.9, signifying its inability to generate counterfactual explanations for 90% of the graphs. This

Table 6: Performance of counterfactual explainers on node classification. Shaded cells indicate the best result in a column. Note that only CF-GNNExplainer and CF$^2$ can explain node classification. In these datasets, ground truth explanations are provided. Hence, accuracy (Acc) represents the percentage of edges within the counterfactual that belong to the ground truth explanation.

| | Tree-Cycles | | | Tree-Grid | | | BA-Shapes | | |
|---|---|---|---|---|---|---|---|---|---|
| Method / Metric | Suffic. ↓ | Size ↓ | Acc.(%) ↑ | Suffic. ↓ | Size ↓ | Acc.(%) ↑ | Suffic. ↓ | Size ↓ | Acc.(%) ↑ |
| **CF-GnnEx** | $0.5 \pm 0.08$ | $1.03 \pm 0.16$ | $100.0 \pm 0.0$ | $0.09 \pm 0.06$ | $1.42 \pm 0.55$ | $92.70 \pm 4.99$ | $0.37 \pm 0.05$ | $1.37 \pm 0.59$ | $91.5 \pm 4.36$ |
| **CF$^2$** ($\alpha = 0$) | $0.76 \pm 0.06$ | $4.55 \pm 1.48$ | $74.71 \pm 18.70$ | $0.99 \pm 0.02$ | $7.0 \pm 0.0$ | $14.29 \pm 0.0$ | $0.25 \pm 0.88$ | $4.24 \pm 1.70$ | $68.89 \pm 12.28$ |

observation suggests that there exists considerable potential for further enhancement. We also note that while CLEAR achieves the best (lowest) sufficiency in AIDS, the number of perturbations it requires (size) is exorbitantly high to be useful in practical use-cases.

**Insights on node classification (Table 6):** We observe that CF-GNNExplainer consistently outperforms CF$^2$ ($\alpha = 0$ indicates the method to be entirely counterfactual). We note that our result contrasts with the reported results in CF$^2$ Tan et al. (2022), where CF$^2$ was shown to outperform CF-GNNExplainer. A closer examination reveals that in Tan et al. (2022), the value of $\alpha$ was set to 0.6, placing a higher emphasis on factual reasoning. It was expected that with $\alpha = 0$, counterfactual reasoning would be enhanced. However, the results do not align with this hypothesis. We note that in CF$^2$, the optimization function is a combination of explanation complexity and explanation strength. The contribution of $\alpha$ is solely in the explanation strength component, based on its alignment with factual and counterfactual reasoning. The counterintuitive behavior observed with $\alpha$ is attributed to the domination of explanation complexity in the objective function, thereby diminishing the intended impact of $\alpha$. Finally, when compared to graph classification, the sufficiency produced by the best methods in the node classification task is significantly lower indicating that it is an easier task. One possible reason might be the space of counterfactuals is smaller in node classification.

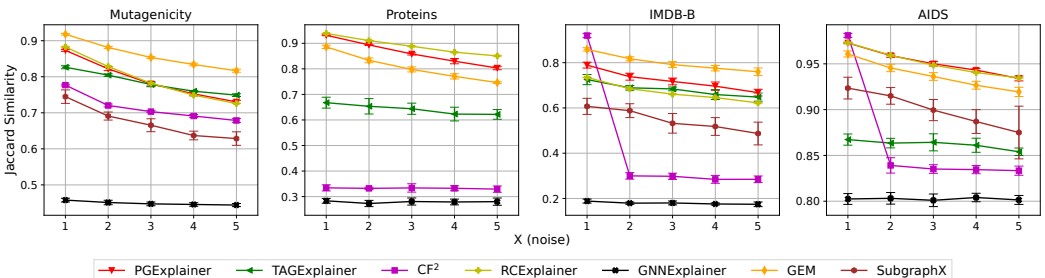

Figure 3: Stability of factual explainers in Jaccard similarity of explanations under topological noise. Here, the $x$-ticks (Noise) denote the number of perturbations made to the edge set of the original graph. Here, perturbations include randomly sampling x(denoted on x axis) negative edges and adding them to the original edge set (i.e., connect a pair of nodes that were previously unconnected).

## 4.2 STABILITY

We next examine the stability of the explanations against topological noise, model parameters, and the choice of GNN architecture. In App. C, we present the impact of the above mentioned factors on other metrics of interest such as sufficiency and explanation size. In addition, we also present impact of feature perturbation and topological adversarial attack in App. C.

**Insights on factual-stability against topological noise:** Fig. 3 illustrates the Jaccard coefficient as a function of the noise volume. Similar to Fig.2, edge selection for the explanation involves a greedy approach that prioritizes the highest score edges. A clear trend that emerges is that inductive methods consistently outperform the transductive methods (such as CF$^2$ and GNNExplainer). This is expected since transductive methods lack generalizable capability to unseen data. Furthermore, the stability is worse on denser datasets of IMDB-B since due to the presence of more edges, the search space of explanation is larger. RCEXPLAINER (executed at $\alpha = 1$) and PGEXPLAINER consistently exhibit higher stability. This consistent performance reinforces the claim that RCEXPLAINER is the preferred factual explainer. The stability of RCEXPLAINER can be attributed to its strategy of selecting a subset of edges that is resistant to changes, such that the removal of these edges

Table 7: Stability in explanations provided by factual explainers across runs. We fix the size to 10 for all explainers. The most stable explainer for each dataset (row) corresponding to the three categories of $1vs2$, $1vs3$ and $2vs3$ are highlighted through gray, yellow and cyan shading respectively.

| Dataset / Seeds | PGExplainer | | | TAGExplainer | | | CF$^2$ | | | RCExplainer | | | GNNExplainer | | |
|---|---|---|---|---|---|---|---|---|---|---|---|---|---|---|---|
| | 1vs2 | 1vs3 | 2vs3 | 1vs2 | 1vs3 | 2vs3 | 1vs2 | 1vs3 | 2vs3 | 1vs2 | 1vs3 | 2vs3 | 1vs2 | 1vs3 | 2vs3 |
| Mutagenicity | 0.69 | 0.75 | 0.62 | 0.76 | 0.78 | 0.74 | 0.77 | 0.77 | 0.77 | 0.75 | 0.71 | 0.71 | 0.46 | 0.47 | 0.47 |
| Proteins | 0.38 | 0.51 | 0.38 | 0.55 | 0.48 | 0.46 | 0.34 | 0.34 | 0.35 | 0.88 | 0.85 | 0.91 | 0.28 | 0.28 | 0.28 |
| Mutag | 0.5 | 0.54 | 0.51 | 0.36 | 0.43 | 0.72 | 0.78 | 0.79 | 0.79 | 0.86 | 0.92 | 0.87 | 0.57 | 0.57 | 0.58 |
| IMDB-B | 0.67 | 0.76 | 0.67 | 0.67 | 0.60 | 0.56 | 0.32 | 0.32 | 0.32 | 0.75 | 0.73 | 0.70 | 0.18 | 0.19 | 0.18 |
| AIDS | 0.88 | 0.87 | 0.82 | 0.81 | 0.83 | 0.87 | 0.85 | 0.85 | 0.85 | 0.95 | 0.96 | 0.97 | 0.80 | 0.80 | 0.80 |
| NCI1 | 0.58 | 0.55 | 0.64 | 0.69 | 0.81 | 0.65 | 0.60 | 0.60 | 0.60 | 0.71 | 0.71 | 0.94 | 0.44 | 0.44 | 0.44 |

Table 8: Stability of factual explainers against the GNN architecture. We fix the size to 10. We report the Jaccard coefficient of explanations obtained for each architecture against the explanation provided over GCN. The best explainers for each dataset (row) are highlighted in gray, yellow and cyan shading for GAT, GIN, and GRAPHSAGE, respectively. GRAPHSAGE is denoted by SAGE.

| Dataset / Architecture | PGExplainer | | | TAGExplainer | | | CF$^2$ | | | RCExplainer | | | GNNExplainer | | |
|---|---|---|---|---|---|---|---|---|---|---|---|---|---|---|---|
| | GAT | GIN | SAGE | GAT | GIN | SAGE | GAT | GIN | SAGE | GAT | GIN | SAGE | GAT | GIN | SAGE |
| Mutagenicity | 0.63 | 0.65 | 0.60 | 0.24 | 0.25 | 0.32 | 0.52 | 0.47 | 0.54 | 0.56 | 0.52 | 0.46 | 0.43 | 0.42 | 0.43 |
| Proteins | 0.22 | 0.47 | 0.38 | 0.45 | 0.41 | 0.18 | 0.28 | 0.28 | 0.28 | 0.37 | 0.41 | 0.42 | 0.28 | 0.28 | 0.28 |
| Mutag | 0.57 | 0.58 | 0.69 | 0.60 | 0.65 | 0.64 | 0.58 | 0.56 | 0.62 | 0.47 | 0.76 | 0.54 | 0.55 | 0.57 | 0.55 |
| IMDB-B | 0.48 | 0.45 | 0.56 | 0.44 | 0.35 | 0.47 | 0.17 | 0.23 | 0.17 | 0.30 | 0.33 | 0.26 | 0.17 | 0.17 | 0.17 |
| AIDS | 0.81 | 0.85 | 0.87 | 0.83 | 0.83 | 0.84 | 0.80 | 0.80 | 0.80 | 0.81 | 0.85 | 0.81 | 0.8 | 0.8 | 0.8 |
| NCI1 | 0.39 | 0.41 | 0.37 | 0.45 | 0.17 | 0.58 | 0.37 | 0.38 | 0.38 | 0.49 | 0.53 | 0.52 | 0.37 | 0.38 | 0.39 |

significantly impacts the prediction made by the remaining graph Bajaj et al. (2021). PGEXPLAINER also incorporates a form of inherent stability within its framework. It builds upon the concept introduced in GNNEXPLAINER through the assumption that the explanatory graph can be modeled as a random Gilbert graph, where the probability distribution of edges is conditionally independent and can be parameterized. This generic assumption holds the potential to enhance the stability of the method. Conversely, TAGEXPLAINER exhibits the lower stability than RCEXPLAINER and PGEXPLAINER, likely due to its reliance solely on gradients in a task-agnostic manner Xie et al. (2022). The exclusive reliance on gradients makes it more susceptible to overfitting, resulting in reduced stability.

**Insights on factual-stability against explainer instances:** Table 7 presents the stability of explanations provided across three different explainer instances on the same black-box GNN. A similar trend is observed, with RCEXPLAINER remaining the most robust method, while GNNEXPLAINER exhibits the least stability. For GNNEXPLAINER, the Jaccard coefficient hovers around $0.5$, indicating significant variance in explaining the same GNN. Although the explanations change, their quality remains stable (as evident from small standard deviation in Fig. 2). This result indicates that multiple explanations of similar quality exist and hence a single explanation fails to complete explain the data signals. This component is further emphasized when we delve into reproducibility (§ 4.3).

**Insights on factual-stability against GNN architectures:** Finally, we explore the stability of explainers across different GNN architectures in Table 8, which has not yet been investigated in the existing literature. For each combination of architectures, we assess the stability by computing the Jaccard coefficient between the explained predictions of the indicated GNN architecture and the default GCN model. One notable finding is that the stability of explainers exhibits a strong correlation with the dataset used. Specifically, in five out of six datasets, the best performing explainer across all architectures is unique. However, it is important to highlight that the Jaccard coefficients across architectures consistently remain low indicating stability against different architectures is the hardest objective due to the variations in their message aggregating schemes.

### 4.3 NECESSITY AND REPRODUCIBILITY

We aim to understand the quality of explanations in terms of necessity and reproducibility. The results are presented in App. D and E. Our findings suggest that necessity is low but increases with the removal of more explanations, while reproducibility experiments reveal that explanations do not provide a comprehensive explanation of the underlying data, and even removing them and retraining the model can produce a similar performance to the original GNN model.

## 4.4 FEASIBILITY

Counterfactual explanations serve as recourses and are expected to generate graphs that adhere to the feasibility constraints of the pertinent domain. We conduct the analysis of feasibility on molecular graphs. It is rare for molecules to be constituted of multiple connected components Vismara & Laurenço (2000). Hence, we study the distribution of molecules that are connected in the original dataset and its comparison to the distribution in counterfactual recourses. We measure the $p$-value of this deviation. App. A.7 presents the results.

## 4.5 VISUALIZATION-BASED ANALYSIS

We include visualizations of the explanations in App. F. Our analysis reveals that a statistically good performance does not always align with human judgment indicating an urgent need for datasets annotated with ground truth explanations. Furthermore, the visualization analysis reinforces the need to incorporate feasibility as a desirable component in counterfactual reasoning.

## 5 CONCLUDING INSIGHTS AND POTENTIAL SOLUTIONS

Our benchmarking study has yielded several insights that can streamline the development of explanation algorithms. We summarize the key findings below (please also see the App. K for our recommendations of explainer for various scenarios).

- **Performance and Stability:** Among the explainers evaluated, RCEXPLAINER consistently outperformed others in terms of efficacy and stability to noise and variational factors (§ 4.1 and § 4.2).
- **Stability Concerns:** Most factual explainers demonstrated significant deviations across explainer instances, vulnerability to topological perturbations and produced significantly different set of explanations across different GNN architectures. These stability notions should therefore be embraced as desirable factors along with other performance metrics.
- **Model Explanation vs. Data Explanation:** Reproducibility experiments (§ 4.3) revealed that retraining with only factual explanations cannot reproduce the predictions fully. Furthermore, even without the factual explanation, the GNN model predicted accurately on the residual graph. This suggests that explainers only capture specific signals learned by the GNN and do not encompass all underlying data signals.
- **Feasibility Issues:** Counterfactual explanations showed deviations in topological distribution from the original graphs, raising feasibility concerns (§ 4.4).

**Potential Solutions:** The aforementioned insights raise important shortcomings that require further investigation. Below, we explore potential avenues of research that could address these limitations.

- **Feasible recourses through counterfactual reasoning:** Current counterfactual explainers predominantly concentrate on identifying the shortest edit path that nudges the graph toward the decision boundary. This design inherently neglects the feasibility of the proposed edits. Therefore, it is imperative to explicitly address feasibility as an objective in the optimization function. One potential solution lies in the vibrant research field of generative modeling for graphs, which has yielded impressive results Goyal et al. (2020); You et al. (2018); Vignac et al. (2023). Generative models, when presented with an input graph, can predict its likelihood of occurrence within a domain defined by a set of training graphs. Integrating generative modeling into counterfactual reasoning by incorporating likelihood of occurrence as an additional objective in the loss function presents a potential remedy.
- **Ante-hoc explanations for stability and reproducibility:** We have observed that if the explanations are removed and the GNN is retrained on the residual graphs, the GNN is often able to recover the correct predictions from our reproducibilty experiments. Furthermore, the explanation exhibit significant instability in the face of minor noise injection. This incompleteness of explainers and instability is likely a manifestation of their *post-hoc* learning framework, wherein the explanations are generated post the completion of GNN training. In this pipeline, the explainers have no visibility to how the GNN would behave to perturbations on the input data, initialization seeds, etc. Potential solutions may lie on moving to an *ante-hoc* paradigm where the GNN and the explainer are jointly trained Kosan et al. (2023); Miao et al. (2022); Fang et al. (2023).

These insights, we believe, open new avenues for advancing GNN explainers, empowering researchers to overcome limitations and elevate the overall quality and interpretability of GNNs.

## 6 ACKNOWLEDGEMENTS

Samidha Verma acknowledges the generous grant received from Microsoft Research India to sponsor her travel to ICLR 2024. Additionally, this project was partially supported by funding from the National Science Foundation under grant #IIS-2229876 and the CSE Research Acceleration Fund of IIT Delhi.

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

APPENDIX

## A  EXPERIMENTAL SETUP

All experiments were conducted using the Ubuntu 18.04 operating system on an NVIDIA DGX Station equipped with four V100 GPU cards, each having 128GB of GPU memory. The system also included 256GB of RAM and a 20-core Intel Xeon E5-2698 v4 2.2 GHz CPU.

The datasets for factual and counterfactual explainers follow an 80:10:10 split for training, validation and testing. We explain some of our design choices below.

- For factual explainers, the inductive explainers are trained on the training data and the reported results are computed on the entire dataset. We also report results only on test data (please see Sec. A.5) comparing only inductive methods. Transductive methods are run on the entire dataset.
- For counterfactual explainers, the inductive explainers are trained on the training data, and the reported results are computed on the test data. Since transductive methods do not have the notion of training and testing separately, they are run only on the test data.

### A.1  BENCHMARK DATASETS

**Datasets for Node classification:** The following datasets have node labels and are used for the node classification task.

- **TREE-CYCLES Ying et al. (2019b):** The base graph used in this dataset is a binary tree, and the motifs consist of **6-node cycles** (Figure D(a)). The motifs are connected to random nodes in the tree. Non-motif nodes are labeled 0, while the motif nodes are labeled 1.
- **TREE-GRID Ying et al. (2019b):** The base graph used in this dataset is a binary tree, and the motif is a $3 \times 3$ **grid** connected to random nodes in the tree (Figure D(b)). Similar to the tree-cycles dataset, the nodes are labeled with binary classes (0 for the non-motif nodes and 1 for the motif nodes).
- **BA-SHAPES Ying et al. (2019b):** The base graph in this dataset is a Barabasi-Albert (BA) graph. The dataset includes **house-shaped** structures composed of 5 nodes (Figure D (c)). Non-motif nodes are assigned class 0, while nodes at the top, middle, and bottom of the motif are assigned classes 1, 2, and 3, respectively.

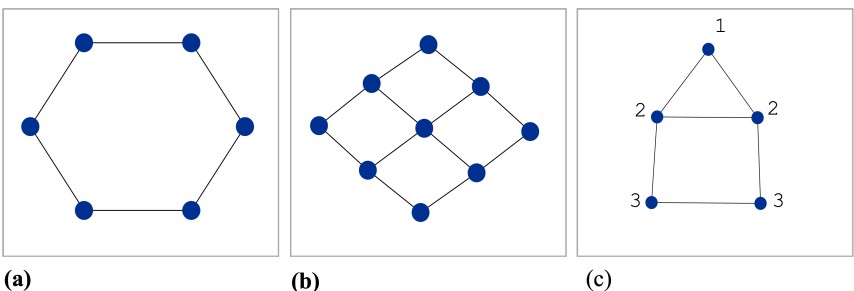

| (a) | (b) | (c) |

Figure D: Motifs used in (a) Tree-Cycles, (b) Tree-Grid and (c) BA-Shapes datasets for the node classification task. Please note the following. (i) Tree-Cycles and Tree-Grid have labels 0 and 1 for the non-motif and the motif nodes, respectively. Hence, all nodes in (a) and (b) have label 1. (ii) BA-Shapes dataset has 4 classes. Non-motif nodes have labels 0; motif nodes have integral labels depending on the position in the house motif. The other labels are 1 (top node), 2 (middle nodes) and 3 (bottom nodes). They are represented in (c).

**Datasets for Graph Classification:**  The following datasets are used for the graph classification task and contain labeled graphs.

- **MUTAG Ivanov et al. (2019) and Mutagenicity Riesen & Bunke (2008); Kazius et al. (2005):** These are graph datasets containing chemical compounds. The nodes represent atoms, and the edges represent chemical bonds. The binary labels depend on the mutagenic effect of the compound

on a bacterium, namely mutagenic or non-mutagenic. MUTAG and Mutagenicity datasets contain 188 and 4337 graphs, respectively.

- **AIDS:** Ivanov et al. (2019) This dataset contains small molecules. The nodes and edges are atoms and chemical bonds, respectively. The molecules are classified by whether they are active against the HIV virus or not.
- **Proteins Borgwardt et al. (2005); Dobson & Doig (2003) and DD Dobson & Doig (2003):** These datasets are comprised of proteins categorized into enzymes and non-enzymes. The nodes represent amino acids, and an edge exists between two nodes if their distance is less than 6 Angstroms.
- **NCI1 Wale et al. (2008):** This dataset is derived from cheminformatics and represents chemical compounds as input graphs. Vertices in the graph correspond to atoms, while edges represent bonds between atoms. This dataset focuses on anti-cancer screenings for cell lung cancer, with chemicals labeled as positive or negative. Each vertex is assigned an input label indicating the atom type, encoded using a one-hot-encoding scheme.
- **IMDB-B Yanardag & Vishwanathan (2015):** The IMDB-BINARY dataset is a collection of movie collaboration networks, encompassing the ego-networks of 1,000 actors and actresses who have portrayed roles in films listed on IMDB. Each network is represented as a graph, where the nodes correspond to the actors/actresses, and an edge is present between two nodes if they have shared the screen in the same movie. These graphs have been constructed specifically from movies in the Action and Romance genres, which are the class labels.
- **REDDIT-B Yanardag & Vishwanathan (2015):** REDDIT-BINARY dataset encompasses graphs representing online discussions on Reddit. Each graph has nodes representing users, connected by edges when either user responds to the other's comment. The four prominent subreddits within this dataset are IAmA, AskReddit, TrollXChromosomes, and atheism. IAmA and AskReddit are question/answer-based communities, while TrollXChromosomes and atheism are discussion-based communities. Graphs are labeled based on their affiliation with either a question/answer-based or discussion-based community.
- **GRAPH-SST2 Yuan et al. (2022):** The Graph-SST2 dataset is a graph-based dataset derived from the SST2 dataset Socher et al. (2013), which contains movie review sentences labeled with positive or negative sentiment. Each sentence in the Graph-SST2 dataset is transformed into a graph representation, with words as nodes and edges representing syntactic relationships capturing the sentence's grammatical structure. The sentiment labels from the original SST2 dataset are preserved, allowing for sentiment analysis tasks using the graph representations of the sentences.
- **ogbg-molhiv** Allamanis et al. (2018): ogbg-molhiv is a molecule dataset with nodes representing atoms and edges representing chemical bonds. The node features represent various properties of the atoms like chirality, atomic number, formal charge etc. Edge attributes represent the bond type. We study binary classification task on this dataset. The task is to achieve the most accurate predictions of specific molecular properties. These properties are framed as binary labels, indicating attributes like whether a molecule demonstrates inhibition of HIV virus replication or not.

## A.2    DETAILS OF GNN MODEL $\Phi$ USED FOR NODE CLASSIFICATION

We use the same GNN model used in CF-GNNEXPLAINER and CF$^2$. Specifically, it is a Graph Convolutional Networks  Kipf & Welling (2016) trained on each of the datasets. Each model has 3 graph convolutional layers with 20 hidden dimensions for the benchmark datasets. The non-linearity used is *relu* for the first two layers and $log$ softmax after the last layer of GCN. The learning rate is 0.01. The train and test data are divided in the ratio 80:20. The accuracy of the GNN model $\Phi$ for each dataset is mentioned in Table I.

Table I: Accuracy of black-box GNN $\Phi$ on the datasets used for node classification, for evaluation of counterfactual explainers. $\Phi$ is a GCN Kipf & Welling (2016) for this task

| Dataset | Train accuracy | Test Accuracy |
|---|---|---|
| Tree-Cycles | 0.9123 | 0.9086 |
| Tree-Grid | 0.8434 | 0.8744 |
| BA-Shapes | 0.9661 | 0.9857 |

## A.3 Details of Base Gnn Model Φ for the Graph Classification Task

Our GNN models have an optional parameter for continuous edge weights, which in our case represents explanations. Each model consists of 3 layers with 20 hidden dimensions specifically designed for benchmark datasets. The models provide node embeddings, graph embeddings, and direct outputs from the model (without any softmax function). The output is obtained through a one-layer MLP applied to the graph embedding. We utilize the max pooling operator to calculate the graph embedding. The dropout rate, learning rate, and batch size are set to 0, 0.001, and 128, respectively. The train, validation, and test datasets are divided into an 80:10:10 ratio. The algorithms run for 1000 epochs with early stopping after 200 patience steps on the validation set. The performance analysis of the base GNN models ( Kipf & Welling (2016); Veličković et al. (2018); Xu et al. (2019); Hamilton et al. (2017)) for each graph classification dataset is presented in Table J.

Table J: Test accuracy of black-box Gnn Φ trained for the graph classification task, averaged over 10 runs with random seeds. We train multiple Gnns for this task to test explainers for stability against Gnn architectures.

| Dataset | GCN | GAT | GIN | GraphSAGE |
|---------|-----|-----|-----|-----------|
| Mutagenicity | $0.8724 \pm 0.0092$ | $0.8685 \pm 0.0111$ | $0.8914 \pm 0.0101$ | $0.8749 \pm 0.0059$ |
| Mutag | $0.925 \pm 0.0414$ | $0.8365 \pm 0.0264$ | $0.9542 \pm 0.0149$ | $0.8323 \pm 0.0445$ |
| Proteins | $0.8418 \pm 0.0144$ | $0.8362 \pm 0.0269$ | $0.8352 \pm 0.0165$ | $0.8408 \pm 0.0124$ |
| IMDB-B | $0.8318 \pm 0.0197$ | $0.8292 \pm 0.015$ | $0.8554 \pm 0.027$ | $0.8373 \pm 0.0093$ |
| AIDS | $0.999 \pm 0.0005$ | $0.9971 \pm 0.0068$ | $0.9797 \pm 0.0099$ | $0.9903 \pm 0.0088$ |
| NCI1 | $0.8243 \pm 0.028$ | $0.8096 \pm 0.015$ | $0.8365 \pm 0.0201$ | $0.8303 \pm 0.0137$ |
| Graph-SST2 | $0.957 \pm 0.001$ | $0.9603 \pm 0.0009$ | $0.9552 \pm 0.0014$ | $0.9611 \pm 0.0011$ |
| DD | $0.736 \pm 0.0377$ | $0.7312 \pm 0.048$ | $0.7693 \pm 0.0238$ | $0.7541 \pm 0.0415$ |
| REDDIT-B | $0.8984 \pm 0.0247$ | $0.8444 \pm 0.0266$ | $0.6886 \pm 0.1231$ | $0.8733 \pm 0.0196$ |
| ogbg-molhiv | $0.9729 \pm 0.0002$ | $0.9722 \pm 0.0010$ | $0.9726 \pm 0.0003$ | $0.9725 \pm 0.0005$ |

## A.4 Details of Factual Explainers for the Graph Classification Task

In many cases, explainers generate continuous explanations that can be used with graph neural network (Gnn) models, which can handle edge weights. To be able to use explanations in our Gnn models, we map them into $[0, 1]$ using a sigmoid function if not mapped. While generating performance results, we calculate top-k edges based on their scores instead of assigning a threshold value (e.g., 0.5). However, there are some approaches, such as GEM and SubgraphX, that do not rely on continuous edge explanations.

GEM employs a variational auto-encoder to reconstruct ground truth explanations. As a result, the generated explanations can include negative values. While our experiments primarily focus on the order of explanations and do not require invoking the base Gnn in the second stage of GEM, we can still use negative explanation edges.

On the other hand, SubgraphX ranks different subgraph explanations based on their scores. We select the top 20 explanations and, for each explanation, compute the subgraph. Then, we enhance the importance of each edge of a particular subgraph by incrementing its score by 1. Finally, we normalize the weights of the edges. This process allows us to obtain continuous explanations as well. Moreover, since SubgraphX employs tree search, its scalability is limited when dealing with large graphs. For instance, in the Mutagenicity dataset, obtaining explanations for 435 graphs requires approximately 26.5 hours. To address this challenge, we restricted our analysis to test graphs when calculating explanations using SubgraphX. It is important to include this disclaimer, working on only subset of graphs may introduce potential biases or noises in the results.

## A.5 Factual Explainers: Inductive Methods on Test Set

The inductive factual explainers are run only on the test data and the results are reported in Figure E. The results are similar to the ones where the methods are run on the entire dataset (Figure 2). Consistent with the earlier results, PGExplainer consistently delivers inferior results compared to other

baseline methods, and no single technique dominates across all datasets. Overall, RCEXPLAINER could be recommended as one of the preferred choices.

Figure E: Sufficiency of the inductive factual explainers against the explanation size on only test data. For factual explanations, higher is better. We omit those methods for a dataset that throw an out-of-memory (OOM) error and are not scalable.

## A.6  CODES AND IMPLEMENTATION

Table K shows the code bases we have used for the explainers. We have adapted the codes based on our base GNN models. Our repository, https://github.com/idea-iitd/gnn-x-bench/, includes the adaptations of the methods to our base models.

Table K: Reference of code repositories.

| Method | Repository |
|---|---|
| PGExplainer Luo et al. (2020) | https://github.com/LarsHoldijk/RE-ParameterizedExplainerForGraphNeuralNetworks/ |
| TAGExplainer Xie et al. (2022) | https://github.com/divelab/DIG/tree/main/dig/xgraph/TAGE/ |
| CF$^2$ Tan et al. (2022) | https://github.com/chrisjtan/gnn_cff |
| RCExplainer Bajaj et al. (2021) | https://developer.huaweicloud.com/develop/aigallery/notebook/detail?id=e41f63d3-e346-4891-bf6a-40e64b4a3278 |
| GNNExplainer Ying et al. (2019b) | https://github.com/LarsHoldijk/RE-ParameterizedExplainerForGraphNeuralNetworks/ |
| GEM Lin et al. (2021b) | https://github.com/wanyu-lin/ICML2021-Gem/ |
| SubgraphX Yuan et al. (2021) | https://github.com/divelab/DIG/tree/main/dig/xgraph/SubgraphX |

## A.7  FEASIBILITY

**Counterfactual explanations:** As shown in Table L, we observe statistically significant deviations from the expected values in two out of three molecular datasets. This suggests a heightened probability of predicting counterfactuals that do not correspond to feasible molecules. This finding underscores

Table L: Assessing the statistical significance of deviations in the number of connected graphs between the test set and their corresponding counterfactual explanations on molecular datasets. Statistically significant deviations with $p$-value$< 0.05$ are highlighted.

| Dataset | RCEXPLAINER | | | $\text{CF}^2$ | | |
|---|---|---|---|---|---|---|
| | Expected Count | Observed Count | $p$-value | Expected Count | Observed Count | $p$-value |
| Mutagenicity | 233.05 | 70 | < 0.00001 | 206.65 | 0 | < 0.00001 |
| Mutag | 11 | 9 | 0.55 | 4 | 1 | 0.13 |
| AIDS | 17.6 | 8 | < 0.00001 | 1.76 | 0 | 0.0001 |

a limitation of counterfactual explainers, which has received limited attention within the research community.

**Factual explanations:** The feasibility metric is commonly used in the context of counterfactual graph explainers because it measures how feasible it is to achieve a specific counterfactual outcome. In other words, it assesses the likelihood of a counterfactual scenario being realized given the constraints and assumptions of the underlying base model. On the other hand, factual explainers aim to explain why a model makes a certain prediction based on the actual input data. They do not involve any counterfactual scenarios, so the feasibility metric is not relevant in this context. Instead, factual explainers may use other metrics such as sufficiency and reproducibility to provide insights into how the model is making its predictions. Therefore, we have not used feasibility metrics for factual explanations.

## B    SUFFICIENCY OF FACTUAL EXPLANATIONS UNDER TOPOLOGICAL NOISE

We check the sufficiency of the factual explanations under noise for four different datasets. Figure F demonstrates the results, including when there is no noise (i.e., when X = 0). We set the explanation size (i.e., the number of edges) to 10 units. We observe that, in most cases, increasing noise results in a decrease in the sufficiency metric for the Mutagenicity and AIDS datasets, which is expected. However, for the Proteins and IMDB-B datasets, even though there are still drops for some methods, others remain stable in sufficiency across different noise levels. This demonstrates that, despite the changes in explanations caused by noise, GNN may still predict the same class under noisy conditions.

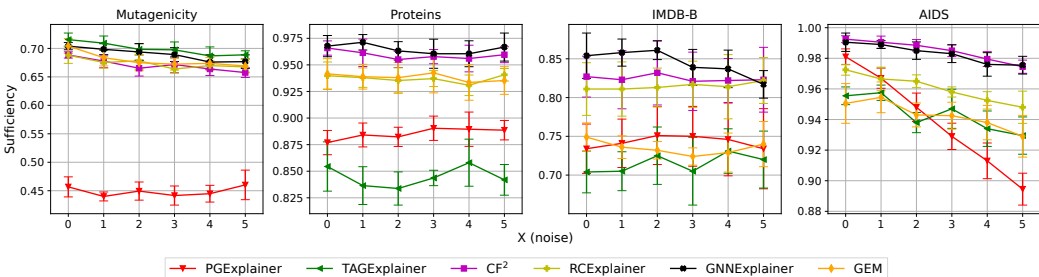

Figure F: Sufficiency of factual explainers under topological noise.

## C    STABILITY

In addition to stability against topological noise, different seeds, and different GNN architecture, we also analyze *stability against feature pertubation* and *stability against topological adversarial attack*.

For feature perturbation, we first select the percentage of nodes to be perturbed ($X\% \in \{10, 20, 30, 40, 50\}$). Then, perturbation operation varies depending on the nature of node features (continuous or discrete). For the Proteins dataset (continuous features); for each feature $f$, we compute its standard deviation $\sigma_f$. Then we sample a value uniformly at random from $\Delta \sim [-0.1, 0.1]$. The feature value $f_x$ is perturbed to $f_x + \Delta \times \sigma_f$. For other datasets (discrete features), for each selected node, we flip its feature to a randomly sampled feature.

For topology adversarial attack, we follow the flip edge method from Wan et al. (2021) with a query size of one and vary the number flip count across datasets.

## C.1 FACTUAL EXPLAINERS

**Stability against feature perturbation:**

Figure G illustrates the outcomes, demonstrating a continuation of the previously observed trends. Among these trends, we see that there is one clear winner for both datasets. However, PGEXPLAINER performs better than most datasets in both datasets (with discrete and continuous features). On the other hand, the transductive method GNNEXPLAINER performs very poorly in both datasets compared to other methods (i.e., inductive), which further provides evidence that transductive methods are poor in stability for factual explanations.

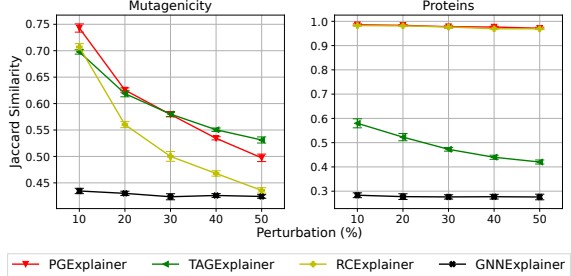

Figure G: Stability of factual explainers against feature perturbation in Jaccard similarity. The stability of explanations drops when the perturbation percentage increases. GNNEXPLAINER (transductive) is the worst method for these two datasets.

**Adversarial attack on topology:** Figure H demonstrates the performance of four factual methods on these evasion attacks for four datasets. The behavior of the factual methods is similar to the topological noise attack explained in Section 4.2 and the feature perturbations results. When an adversarial attack is compared to random perturbations (Fig. 3), we observe higher deterioration in stability, which is expected since adversarial edge flip attack aims every possible edge in the graph rather than only considering nonexistent edges. Similar to feature perturbation, GNNExplainer (transductive) is affected more by the adversarial attack.

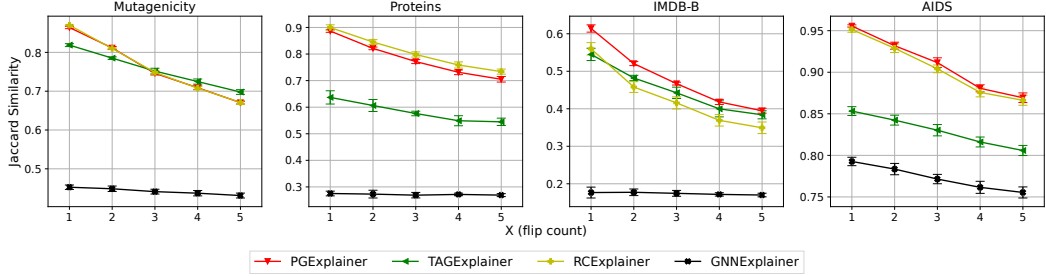

Figure H: Stability of factual explainers against random edge flip in Jaccard similarity. The stability of explanations drops when the flip count increases. GNNEXPLAINER (transductive) is the worst method for these two datasets.

## C.2 COUNTERFACTUAL EXPLAINERS

**Stability against topological noise:** In this section, we investigate the influence of topological noise on datasets on both the performance and generated explanations of counterfactual explainers. For inductive methods (RCEXPLAINER and CLEAR), we utilize explainers trained on noise-free data and only infer on the noisy data. However, for the transductive method $CF^2$, we retrain the model using the noisy data.

Figure I presents the average Jaccard similarity results, indicating the similarity between the counterfactual graph predicted as an explanation for the original graph and the noisy graphs at varying levels of perturbations. Additionally, Figure J demonstrates the performance of different explainers in terms of sufficiency and size as the degree of noise increases. This provides insights into how these explainers handle higher levels of noise.

RCExplainer outperforms other baselines by a significant margin in terms of size and sufficiency across datasets, as shown in Fig. J. However, the Jaccard similarity between RCExplainer and $CF^2$ for counterfactual graphs is nearly identical, as shown in Fig. I. $CF^2$ benefits from its transductive training on noisy graphs. CLEAR's results are not shown for Proteins and Mutagenicity datasets due to scalability issues. In the case of IMDB-B dataset, CLEAR is highly unstable in predicting counterfactual graphs, indicated by a low Jaccard index (Fig. I). Additionally, CLEAR demonstrates high sufficiency but requires a large number of edits, indicating difficulty in finding minimal-edit counterfactuals (Fig. J).

Overall, RCExplainer seems to be the model of choice when topological noise is introduced, and it is significantly faster than $CF^2$ because it is inductive. Further, it is better than CLEAR as the latter does not scale for larger datasets and is inferior in terms of sufficiency and size as well.

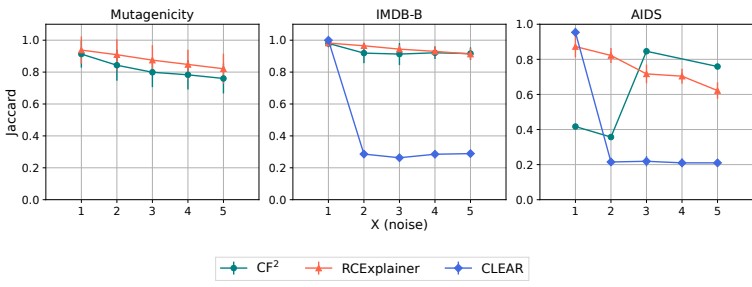

Figure I: Stability of counterfactual explainers against topological noise (Jaccard). We omit CLEAR for Mutagenicity and Proteins as it throws an OOM error for these datasets. The absence of markers representing $CF^2$ in the Protein dataset's plot indicates that counterfactual graphs were not predicted at the corresponding noise values by the method. Overall, RCExplainer performs best in terms of the Jaccard index.

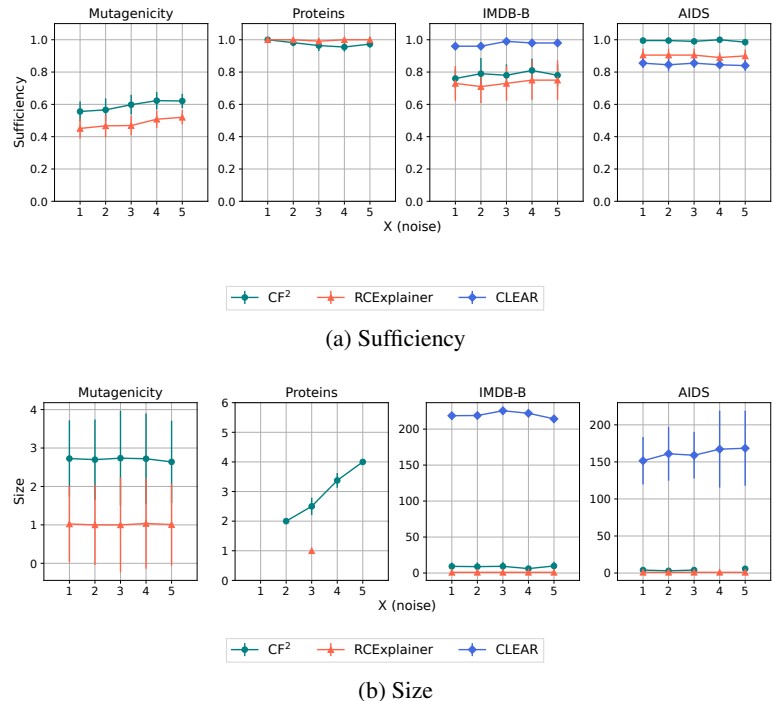

(a) Sufficiency

(b) Size

Figure J: Performance evaluation of counterfactual explainers against topological noise. We omit CLEAR for Mutagenicity and Proteins as it throws an OOM error for these datasets. RCExplainer is more robust to noise in both metrics: (a) sufficiency and (b) size.

**Stability against explainer instances:** Table M provides an overview of the stability exhibited among explainer instances trained using three distinct seeds. Notably, we observe a substantial Jaccard index, indicating favorable stability, in the case of RCEXPLAINER and $CF^2$ explainers. Conversely, CLEAR fails to demonstrate comparable stability. These findings align with the outcomes derived from Table 5. Specifically, when RCEXPLAINER and $CF^2$ are successful in identifying a counterfactual, the resultant counterfactual graphs are obtained through a small number of perturbations. Consequently, the counterfactual graphs exhibit similarities to the original graph, rendering them akin to one another. However, this trend does not hold for CLEAR, as it necessitates a significantly greater number of perturbations.

Table M: Stability against explainer instances. Note that stability with respect to a graph is computable only if both explainer instances find their counterfactual. "NA" indicates no such graph exists.

| Dataset / Seeds | RCExplainer | | | $CF^2$ | | | CLEAR | | |
|---|---|---|---|---|---|---|---|---|---|
| | 1vs2 | 1vs3 | 2vs3 | 1vs2 | 1vs3 | 2vs3 | 1vs2 | 1vs3 | 2vs3 |
| Mutagenicity | 0.96 ±0.06 | 0.96 ±0.04 | 0.98 ±0.03 | 0.90 ±0.09 | 0.89 ±0.1 | 0.89 ±0.11 | OOM | OOM | OOM |
| Proteins | 0.95 ±0.0 | 0.94 ±0.0 | 0.90 ±0.0 | NA | NA | NA | OOM | OOM | OOM |
| Mutag | 0.98 ±0.03 | 0.98 ±0.03 | 1.0 ±0.0 | 1.0 ±0.0 | 1.0 ±0.0 | 1.0 ±0.0 | 0.55 ±0.01 | 0.53 ±0.01 | 0.54 ±0.02 |
| IMDB-B | 0.99 ±0.01 | 1.0 ±0.0 | 0.99 ±0.01 | 0.96 ±0.05 | 0.95 ±0.06 | 0.94 ±0.07 | 0.28 ±0.0 | 0.27 ±0.0 | 0.28 ±0.0 |
| AIDS | 0.84 ±0.04 | 0.96 ±0.06 | 0.84 ±0.04 | NA | 1.0 ±0.0 | NA | 0.19 ±0.02 | 0.20 ±0.03 | 0.19 ±0.04 |
| ogbg-molhiv | 0.99 ±0.03 | 0.99 ±0.04 | 0.99 ±0.04 | 0.801 ±0.149 | 0.764 ±0.14 | 0.784 ±0.144 | OOM | OOM | OOM |

Similarly, as additional results, we also present the variations in terms of sufficiency (Table N) and the explanation size (Table O) produced by the methods using three different seeds. In terms of sufficiency, the methods show stability while varying the seeds. However, the results are drastically different for explanation size. Table O present the results. We observe that RCEXPLAINER is consistently the most stable method while $CF^2$ is worse. The worst stability is shown by CLEAR and this observation is consistent with the previous results.

Table N: Stability of sufficiency produced by counterfactual explainers against the explainer instances (seeds). The best explainers for each dataset (row) are highlighted in gray, yellow and cyan shading for seeds 1, 2, and 3, respectively. OOM indicates that the explainer threw an out-of-memory error.

| Dataset / Seeds | RCExplainer | | | $CF^2$ | | | CLEAR | | |
|---|---|---|---|---|---|---|---|---|---|
| | 1 | 2 | 3 | 1 | 2 | 3 | 1 | 2 | 3 |
| Mutagenicity | 0.4 ±0.06 | 0.40 ±0.05 | 0.41 ±0.05 | 0.50 ±0.05 | 0.49 ±0.06 | 0.52 ±0.05 | OOM | OOM | OOM |
| Proteins | 0.96 ±0.02 | 0.96 ±0.02 | 0.96 ±0.02 | 1.0 ±0.0 | 1.0 ±0.0 | 0.98 ±0.02 | OOM | OOM | OOM |
| Mutag | 0.4 ±0.12 | 0.6 ±0.12 | 0.55 ±0.1 | 0.9 ±0.12 | 0.85 ±0.2 | 0.9 ±0.12 | 0.55 ±0.1 | 0.55 ±0.1 | 0.65 ±0.12 |
| IMDB-B | 0.72 ±0.11 | 0.72 ±0.11 | 0.72 ±0.11 | 0.81 ±0.07 | 0.82 ±0.08 | 0.81 ±0.07 | 0.96 ±0.02 | 0.96 ±0.02 | 0.96 ±0.02 |
| AIDS | 0.91 ±0.04 | 0.91 ±0.04 | 0.91 ±0.04 | 0.98 ±0.02 | 1.0 ±0.0 | 0.99 ±0.01 | 0.84 ±0.03 | 0.82 ±0.03 | 0.83 ±0.04 |
| ogbg-molhiv | 0.90 ±0.02 | 0.88 ±0.01 | 0.90 ±0.01 | 0.96 ±0.01 | 0.97 ±0.00 | 0.96 ±0.00 | OOM | OOM | OOM |

Table O: Stability of *explanation size* produced by explainers against the explainer instances (seeds). NA indicates the inability to find a counterfactual. OOM indicates that the explainer threw out-of-memory error. The best explainers for each dataset (row) are highlighted in gray, yellow and cyan shading for seeds 1, 2, and 3, respectively.

| Dataset / Seeds | RCExplainer | | | $CF^2$ | | | CLEAR | | |
|---|---|---|---|---|---|---|---|---|---|
| | 1 | 2 | 3 | 1 | 2 | 3 | 1 | 2 | 3 |
| Mutagenicity | 1.01 ±0.19 | 1.0 ±0.0 | 1.25 ±0.0 | 2.78 ±0.98 | 2.85 ±1.07 | 2.95 ±1.37 | OOM | OOM | OOM |
| Proteins | 1.0 ±0.0 | 1.0 ±0.0 | 1.0 ±0.0 | NA | NA | 3.0 ±0.0 | OOM | OOM | OOM |
| Mutag | 1.1 ±0.22 | 1.0 ±0.0 | 1.0 ±0.0 | 1.0 ±0.0 | 1.25 ±0.35 | 1.0 ±0.0 | 17.15 ±1.62 | 15.6 ±1.86 | 19.05 ±1.31 |
| IMDB-B | 1.0 ±0.0 | 1.0 ±0.0 | 1.0 ±0.0 | 8.57 ±4.99 | 8.29 ±4.50 | 9.01 ±5.58 | 218.62 ±0.0 | 182.25 ±0.0 | 181.38 ±0.0 |
| AIDS | 1.0 ±0.0 | 1.0 ±0.0 | 1.0 ±0.0 | 5.25 ±0.35 | NA | 6.0 ±0.0 | 164.95 ±47.93 | 162.32 ±45.70 | 185.29 ±78.92 |
| ogbg-molhiv | 1.0 ±0.0 | 1.0 ±0.0 | 1.02 ±0.42 | 10.45 ±4.43 | 9.69 ±4.18 | 10.24 ±4.87 | OOM | OOM | OOM |

**Stability against GNN architectures:** Table P shows the stability of the explainers across different GNN architectures. Similar to our factual setting (Table 8), we assess the stability by computing the Jaccard coefficient between the explained predictions of the indicated GNN architecture and the default GCN model. Unsurprisingly, the stability of the explainer highly depends on the dataset.

RCEXPLAINER is also the most stable among all the explainers, and the produced high values indicate that the method is agnostic towards the variations in different message aggregating schemes of the architectures.

We further look into the stability of the counterfactual methods in terms of sufficiency (Table Q) and the explanation size (Table R) across different GNN architectures. The sufficiency results (Table Q) show large variations produced by the same method on the same dataset due to the different architectures and message passing schemes. For instance, RCEXPLAINER produces sufficiency of .10 and .93 on the AIDS dataset for GAT and GIN, respectively. In terms of explanation size(Table R), RCEXPLAINER is stable against different GNN architectures. However, consistent with previous stability results, $CF^2$ is more unstable than RCEXPLAINER and the worst stability is shown by CLEAR.

Table P: Stability of counterfactual explainers against the GNN architecture. We report the Jaccard coefficient of explanations obtained for GAT, GIN and GRAPHSAGE against the explanation provided over GCN. The higher the Jaccard, the more is the stability. The best explained for each dataset (row) are highlighted in gray, yellow and cyan shading for architectures GAT, GIN, and GRAPHSAGE, respectively. GRAPHSAGE is denoted by SAGE. NA indicates one or both of the architectures were unable to identify a counterfactual for the graphs. OOM indicates that the explainer threw an out-of-memory error.

| Dataset / Architecture | RCEXplainer | | | $CF^2$ | | | CLEAR | | |
|---|---|---|---|---|---|---|---|---|---|
| | GAT | GIN | SAGE | GAT | GIN | SAGE | GAT | GIN | SAGE |
| Mutagenicity | 0.95 ±0.05 | 0.94 ±0.06 | 0.95 ±0.03 | 0.79 ±0.13 | 0.75 ±0.16 | 0.84 ±0.10 | OOM | OOM | OOM |
| Proteins | 0.88 ±0.0 | NA | 0.88 ±0.0 | NA | NA | NA | OOM | OOM | OOM |
| Mutag | 0.94 ±0.0 | NA | 0.90 ±0.02 | NA | NA | NA | 0.86 ±0.0 | NA | 0.72 ±0.04 |
| IMDB-B | 0.99 ±0.01 | 0.98 ±0.0 | 0.98 ±0.01 | NA | 0.93 ±0.0 | NA | 0.60 ±0.0 | 0.70 ±0.0 | 0.76 ±0.0 |
| AIDS | 0.89 ±0.03 | NA | NA | 0.74 ±0.0 | 0.73 ±0.11 | 0.72 ±0.12 | 0.25 ±0.04 | 0.54 ±0.04 | 0.66 ±0.04 |
| ogbg-molhiv | 0.96 ±0.02 | 0.96 ±0.01 | 0.96 ±0.02 | 0.63 ±0.12 | 0.13 ±0.14 | 0.61 ±0.16 | OOM | OOM | OOM |

Table Q: Stability in terms of *sufficiency* of counterfactual explainers against the GNN architectures. OOM indicates that the explainer threw out-of-memory error. The best explainers for each dataset (row) are highlighted in gray, yellow, cyan, and pink shading for GCN, GAT, GIN, SAGE, respectively. RCExplainer outperforms other baselines on a majority of the datasets and architectures. CLEAR also is stable in terms of sufficiency but has a much larger explanation size compared to other baselines(Refer Table R).

| Dataset / Architecture | RCEXplainer | | | | $CF^2$ | | | | CLEAR | | | |
|---|---|---|---|---|---|---|---|---|---|---|---|---|
| | GCN | GAT | GIN | SAGE | GCN | GAT | GIN | SAGE | GCN | GAT | GIN | SAGE |
| Mutagenicity | 0.4 ±0.06 | 0.38 ±0.04 | 0.6 ±0.04 | 0.59 ±0.06 | 0.50 ±0.05 | 0.64 ±0.04 | 0.57 ±0.08 | 0.62 ±0.03 | OOM | OOM | OOM | OOM |
| Proteins | 0.96 ±0.02 | 0.88 ±0.04 | 0.3 ±0.05 | 0.46 ±0.08 | 1.0 ±0.0 | 1.0 ±0.0 | 0.76 ±0.06 | 0.79 ±0.02 | OOM | OOM | OOM | OOM |
| Mutag | 0.4 ±0.12 | 0.7 ±0.19 | 1.0 ±0.0 | 0.45 ±0.19 | 0.9 ±0.12 | 0.9 ±0.12 | 0.45 ±0.33 | 0.7 ±0.19 | 0.55 ±0.1 | 1.0 ±0.0 | 1.0 ±0.0 | 0.05 ±0.1 |
| IMDB-B | 0.72 ±0.11 | 0.89 ±0.02 | 0.54 ±0.06 | 0.39 ±0.04 | 0.81 ±0.07 | 1.0 ±0.0 | 0.98 ±0.02 | 0.99 ±0.02 | 0.96 ±0.02 | 0.68 ±0.08 | 0.22 ±0.11 | 0.32 ±0.11 |
| AIDS | 0.91 ±0.04 | 0.10 ±0.04 | 0.93 ±0.03 | 0.86 ±0.05 | 0.98 ±0.02 | 0.92 ±0.04 | 0.96 ±0.01 | 0.96 ±0.02 | 0.84 ±0.03 | 0.80 ±0.04 | 0.74 ±0.04 | 0.84 ±0.02 |
| ogbg-molhiv | 0.90 ±0.02 | 0.80 ±0.01 | 0.56 ±0.01 | 0.20 ±0.01 | 0.96 ±0.01 | 0.96 ±0.01 | 0.90 ±0.01 | 0.59 ±0.01 | OOM | OOM | OOM | OOM |

Table R: Stability of *explanation size* produced by explainers against different GNN architectures. OOM indicates that the explainer threw out-of-memory error. NA indicates that the explainer could not identify a counterfactual for the graphs. The best explainers for each dataset (row) are highlighted in gray, yellow, cyan, and pink shading for GCN, GAT, GIN, SAGE respectively. RCExplainer outperforms other counterfactual baselines.

| Dataset / Architecture | RCEXplainer | | | | $CF^2$ | | | | CLEAR | | | |
|---|---|---|---|---|---|---|---|---|---|---|---|---|
| | GCN | GAT | GIN | SAGE | GCN | GAT | GIN | SAGE | GCN | GAT | GIN | SAGE |
| Mutagenicity | 1.01 ± 0.18 | 1.33 ± 2.06 | 1.0 ± 0.0 | 1.03 ± 0.29 | 2.81 ± 1.12 | 3.90 ± 2.08 | 5.93 ± 2.97 | 3.32 ± 1.61 | OOM | OOM | OOM | OOM |
| Proteins | 1.0 ± 0.0 | 1.0 ± 0.0 | 1.0 ± 0.0 | 1.97 ± 7.75 | 3.5 ± 0.5 | 4.0 ± 0.0 | 2.62 ± 1.22 | 2.04 ± 1.3 | OOM | OOM | OOM | OOM |
| Mutag | 1.0 ± 0.0 | NA | NA | 1.0 ± 0.0 | 2.0 ± 1.07 | 1.0 ± 0.0 | 20.36 ± 3.94 | 1.4 ± 0.8 | 38.12 ± 3.41 | 37.4 ± 3.61 | NA | 45.76 ± 7.94 |
| IMDB-B | 1.0 ± 0.0 | 1.0 ± 0.0 | 1.0 ± 0.0 | 1.0 ± 0.0 | 7.78 ± 3.98 | NA | 6.0 ± 0.0 | 7.17 ± 3.89 | 424.0 ± 192.26 | 441.53 ± 70.97 | 475.42 ± 75.76 | 350.45 ± 86.62 |
| AIDS | 1.0 ± 0.0 | 1.0 ± 0.0 | 1.0 ± 0.0 | 1.0 ± 0.0 | NA | 4.21 ± 3.07 | 2.0 ± 0.0 | 3.22 ± 2.15 | 222.84 ± 47.02 | 667.01 ± 94.35 | 212.77 ± 43.18 | 201.09 ± 38.85 |

**Stability to feature noise:** Table S presents the impact of feature noise on counterfactual explanations in Mutag and Mutagenicity. We observe that $CF^2$ and CLEAR are markedly more stable than

RCEXPLAINER in Mutag. This outcome is not surprising, considering that RCEXPLAINER exclusively addresses topological perturbations, while both $CF^2$ and CLEAR accommodate perturbations encompassing both topology and features. In Mutaganecity, RCEXPLAINER exhibits slightly higher stability than $CF^2$.

Table S: Stability of counterfactual explainers against feature perturbation on "Mutag" and "Mutagenecity" datasets. We do not report results for CLEAR on Mutagenicity since it runs out of GPU memory.

(a) Mutag

| | RCExplainer | | | CF$^2$ | | | CLEAR | | |
|---|---|---|---|---|---|---|---|---|---|
| Noise% / Metric | Sufficiency | Size | Jaccard | Sufficiency | Size | Jaccard | Sufficiency | Size | Jaccard |
| 0 (no noise) | $0.4 \pm 0.12$ | $1.10 \pm 0.22$ | $1.0 \pm 0.0$ | $0.90 \pm 0.12$ | $1.0 \pm 0.0$ | $1.0 \pm 0.0$ | $0.55 \pm 0.1$ | $17.15 \pm 1.62$ | $1.0 \pm 0.0$ |
| 10 | $1.0 \pm 0.0$ | NA | NA | $0.6 \pm 0.2$ | $2.17 \pm 0.31$ | $0.19 \pm 0.0$ | $0.55 \pm 0.1$ | $16.35 \pm 1.68$ | $0.98 \pm 0.01$ |
| 20 | $0.75 \pm 0.22$ | $1.0 \pm 0.0$ | NA | $0.25 \pm 0.16$ | $1.95 \pm 0.80$ | NA | $0.55 \pm 0.1$ | $18.1 \pm 1.94$ | $0.55 \pm 0.01$ |
| 30 | $1.0 \pm 0.0$ | NA | NA | $0.6 \pm 0.3$ | $1.0 \pm 0.0$ | $0.29 \pm 0.0$ | $0.55 \pm 0.1$ | $19.1 \pm 2.91$ | $0.57 \pm 0.02$ |
| 40 | $1.0 \pm 0.0$ | NA | NA | $0.6 \pm 0.2$ | $2.8 \pm 0.0$ | $0.29 \pm 0.0$ | $0.55 \pm 0.1$ | $14.7 \pm 1.78$ | $0.58 \pm 0.02$ |
| 50 | $1.0 \pm 0.0$ | NA | NA | $0.8 \pm 0.1$ | $1.5 \pm 0.0$ | NA | $0.55 \pm 0.1$ | $16.05 \pm 2.48$ | $0.54 \pm 0.02$ |

(b) Mutagenicity

| | RCEXplainer | | | CF$^2$ | | |
|---|---|---|---|---|---|---|
| Noise% / Metric | Sufficiency | Size | Jaccard | Sufficiency | Size | Jaccard |
| 0 (no noise) | $0.4 \pm 0.06$ | $1.01 \pm 0.19$ | $1.0 \pm 0.0$ | $0.50 \pm 0.05$ | $2.78 \pm 0.98$ | $1.0 \pm 0.0$ |
| 10 | $0.43 \pm 0.04$ | $1.01 \pm 0.19$ | $0.96 \pm 0.04$ | $0.49 \pm 0.04$ | $2.11 \pm 1.06$ | $0.92 \pm 0.06$ |
| 20 | $0.47 \pm 0.06$ | $1.0 \pm 0.0$ | $0.95 \pm 0.04$ | $0.53 \pm 0.06$ | $1.73 \pm 0.94$ | $0.86 \pm 0.08$ |
| 30 | $0.50 \pm 0.04$ | $1.0 \pm 0.0$ | $0.94 \pm 0.04$ | $0.61 \pm 0.04$ | $1.68 \pm 0.91$ | $0.86 \pm 0.09$ |
| 40 | $0.52 \pm 0.05$ | $1.0 \pm 0.0$ | $0.93 \pm 0.06$ | $0.54 \pm 0.05$ | $1.47 \pm 0.73$ | $0.87 \pm 0.09$ |
| 50 | $0.49 \pm 0.05$ | $1.0 \pm 0.0$ | $0.93 \pm 0.06$ | $0.67 \pm 0.01$ | $1.54 \pm 1.06$ | $0.86 \pm 0.08$ |

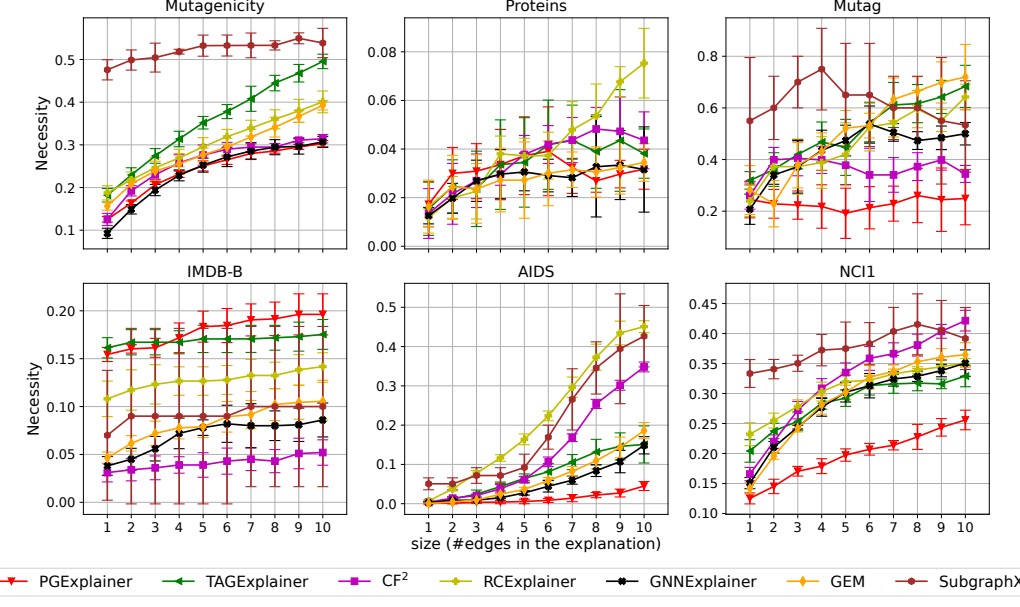

Figure K: Necessity of various factual explainers against the explanation size. The necessity increases with the removal of the explanations.

## D   NECESSITY: RESULTS FOR SEC. 4.3

For necessity, we remove explanations from graphs and measure the ratio of graphs for which the GNN prediction on the residual graph is flipped. We expect that removing more explanations will lead to more necessity; however, we do not necessarily expect necessity to be higher because our factual explainers are not trained to make the residual graph counterfactual. Fig. K presents the necessity performance on six datasets for all factual methods with varying explanation sizes from 1 to 10. The trend aligns with our expectations, as the removal of more explanations increases the necessity. The value of necessity varies between datasets. Proteins and IMDB-B datasets have graphs with larger sizes (in terms of the number of edges), which aligns with the small necessity score. On the other hand, datasets with relatively smaller graphs have higher necessity scores. Note that our factual methods do not optimize residual graphs to be counterfactuals; this might be another reason for the low values.

## E   REPRODUCIBILITY: RESULTS FOR SEC. 4.3

Reproducibility can be measured two different ways. (1) Retraining using only explanation graphs called Reproducibility$^+$, (2) retraining using only residual graphs called Reproducibility$^-$. Both metrics is a ratio of an GNN accuracy compared to the original GNN accuracy. We provide the math definitions in Table 3. In our figures, we separated SubgraphX to an independent table, because we could only obtain explanations of test graphs for SubgraphX (refer to our disclaimer in Sec. A.4)

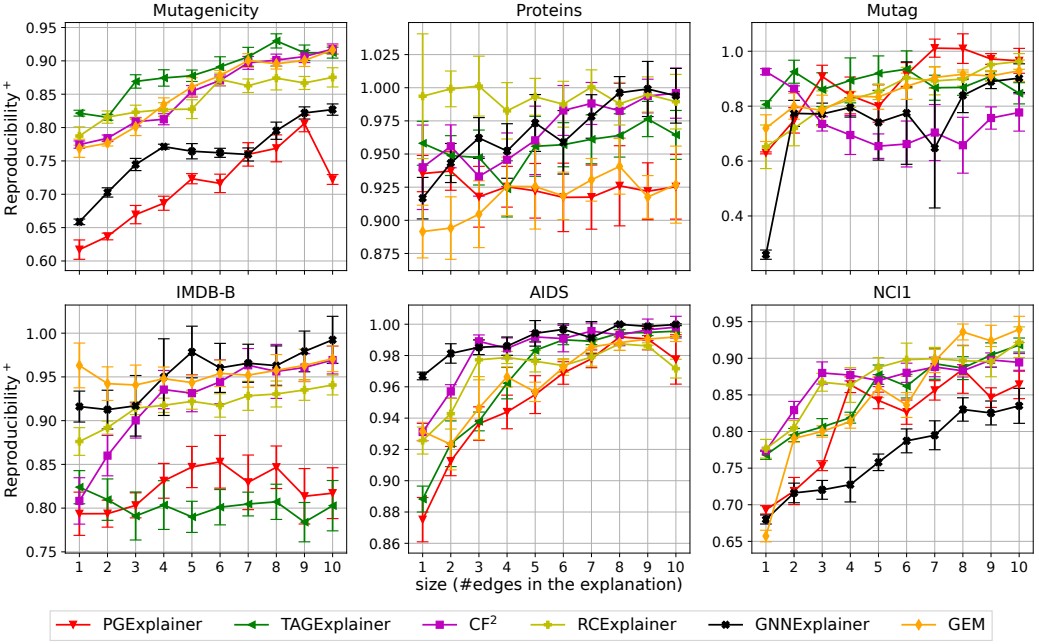

Figure L: Reproducibility$^+$ of factual explainers against size. Performance tends to rise with more edges in the explanations; however, a small number of edges does not guarantee a performance of 1.0.

| Datasets | Size (#edges in the explanation) | | | | | | | | | |
|---|---|---|---|---|---|---|---|---|---|---|
| | 1 | 2 | 3 | 4 | 5 | 6 | 7 | 8 | 9 | 10 |
| Mutagenicity | $1.11 \pm 0.02$ | $1.07 \pm 0.03$ | $1.08 \pm 0.02$ | $1.05 \pm 0.02$ | $1.05 \pm 0.02$ | $1.05 \pm 0.02$ | $1.03 \pm 0.02$ | $1.01 \pm 0.02$ | $0.97 \pm 0.02$ | $1.02 \pm 0.02$ |
| Mutag | $0.86 \pm 0.43$ | $0.43 \pm 0.53$ | $0.76 \pm 0.5$ | $1.08 \pm 0.0$ | $0.11 \pm 0.32$ | $0.22 \pm 0.43$ | $1.08 \pm 0.0$ | $0.86 \pm 0.43$ | $0.97 \pm 0.32$ | $1.08 \pm 0.0$ |
| IMDB-B | $1.07 \pm 0.16$ | $1.1 \pm 0.17$ | $1.05 \pm 0.15$ | $1.08 \pm 0.06$ | $1.09 \pm 0.08$ | $1.07 \pm 0.08$ | $1.05 \pm 0.06$ | $1.06 \pm 0.06$ | $1.02 \pm 0.05$ | $1.05 \pm 0.09$ |
| AIDS | $1.0 \pm 0.0$ | $1.0 \pm 0.0$ | $1.0 \pm 0.0$ | $1.0 \pm 0.0$ | $1.0 \pm 0.01$ | $1.0 \pm 0.01$ | $0.99 \pm 0.01$ | $1.0 \pm 0.0$ | $1.0 \pm 0.0$ | $1.0 \pm 0.0$ |
| NCI1 | $1.05 \pm 0.04$ | $1.09 \pm 0.03$ | $1.09 \pm 0.03$ | $1.08 \pm 0.02$ | $1.09 \pm 0.02$ | $1.09 \pm 0.02$ | $1.07 \pm 0.03$ | $1.07 \pm 0.03$ | $1.05 \pm 0.03$ | $1.04 \pm 0.04$ |

Table T: Reproducibility$^+$ in SubgraphX. Since we use small number of graphs for SubgraphX, the variance of the results are very high, thus unreliable.

Figure L and Table T illustrate the Reproducibility$^+$ performance of seven factual methods against the size of explanations for six datasets. Reproducibility increases with more edges in the explanations for the most cases, as expected. However, reaching a score of 1.0 is challenging even when selecting the most crucial edges. This suggests that explanations do not capture the full picture for GNN predictions.

Figure M and Table U illustrate the Reproducibility$^-$ performance of seven factual methods against the size of explanations for six datasets. Reproducibility remains high even when the most crucial edges from the graphs are removed. This demonstrates that the explainers hardly capture the real cause of the GNN predictions.

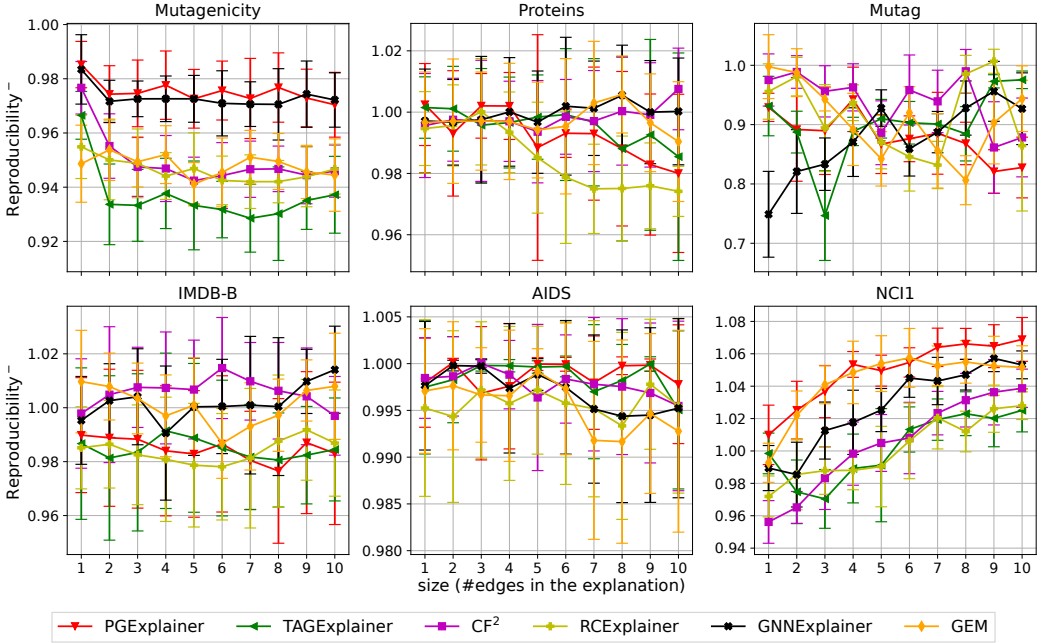

Figure M: Reproducibility$^-$ of factual explainers against size. Even when the most crucial edges are taken out, the performance still remains close to 1.0.

| | Size (#edges in the explanation) | | | | | | | | | |
|---|---|---|---|---|---|---|---|---|---|---|
| Datasets | 1 | 2 | 3 | 4 | 5 | 6 | 7 | 8 | 9 | 10 |
| Mutagenicity | $1.06 \pm 0.03$ | $1.01 \pm 0.02$ | $1.01 \pm 0.02$ | $1.01 \pm 0.03$ | $1.0 \pm 0.03$ | $1.0 \pm 0.03$ | $1.01 \pm 0.03$ | $0.99 \pm 0.03$ | $1.0 \pm 0.03$ | $0.99 \pm 0.03$ |
| Mutag | $1.08 \pm 0.0$ | $1.08 \pm 0.0$ | $0.97 \pm 0.32$ | $1.08 \pm 0.0$ | $0.97 \pm 0.32$ | $0.97 \pm 0.32$ | $1.08 \pm 0.0$ | $1.08 \pm 0.0$ | $1.08 \pm 0.0$ | $1.08 \pm 0.0$ |
| IMDB-B | $0.82 \pm 0.28$ | $0.96 \pm 0.14$ | $0.86 \pm 0.27$ | $0.85 \pm 0.28$ | $0.86 \pm 0.28$ | $0.82 \pm 0.28$ | $0.84 \pm 0.27$ | $0.85 \pm 0.28$ | $0.85 \pm 0.28$ | $0.86 \pm 0.28$ |
| AIDS | $1.0 \pm 0.0$ | $1.0 \pm 0.01$ | $1.0 \pm 0.01$ | $1.0 \pm 0.0$ | $1.0 \pm 0.0$ | $1.0 \pm 0.01$ | $0.99 \pm 0.01$ | $1.0 \pm 0.01$ | $1.0 \pm 0.01$ | $1.0 \pm 0.01$ |
| NCI1 | $0.9 \pm 0.06$ | $0.91 \pm 0.05$ | $0.89 \pm 0.07$ | $0.93 \pm 0.07$ | $0.92 \pm 0.07$ | $0.92 \pm 0.06$ | $0.91 \pm 0.05$ | $0.9 \pm 0.04$ | $0.91 \pm 0.05$ | $0.92 \pm 0.05$ |

Table U: Reproducibility$^-$ in SubgraphX. Since we use small number of graphs for SubgraphX, the variance of the results are very high, thus unreliable.

## F    VISUALIZATION OF EXPLANATIONS

In Figs. N and  O, we engage in a visual analysis of the explanations provided by various GNN explainers. The graphs presented in these figures represent mutagenic molecules sourced from the Mutag dataset. Several insights emerge from this analysis.

**Factual:** The mutagenic attribute of the molecule in Fig. N stems from the presence of the $NO_2$ group attached to the benzene ringYing et al. (2019b); Debnath et al. (1991). As a result, the optimal explanation entails pinpointing this specific benzene ring in conjunction with the $NO_2$ group. Notably, we observe that while certain explainers identify fragments of this subgraph, with $CF^2$ achieving the highest overlap, many also highlight bonds originating from regions outside the authentic explanatory context. Adding to the intrigue, the explanation offered by RCEXPLAINER stands out due to its compactness, resulting in commendable statistical performance. However, this succinct explanation

lacks meaning in the eyes of a domain expert. Consequently, a pressing need arises for real-world datasets endowed with ground truth explanations, a resource that the current field unfortunately lacks.

**Counterfactuals:** Fig. O illustrates two molecules, with Molecule 1 (top row) being identical to the one shown in Fig. N. The optimal explanation involves eliminating the $NO_2$ component, a task accomplished solely by $CF^2$ in the case of Molecule 1. While the remaining explanation methods can indeed alter the GNN prediction by implementing the changes described in Figure O, two critical insights emerge. First, statistically, RCEXPLAINER is considered a better explanation than $CF^2$ since its size is 1 compared to 3 of $CF^2$. However, our interaction with multiple chemists clearly indicated their preference towards $CF^2$ since eliminates the entire $NO_2$ group. Second, chemically infeasible explanations are common as evident from CLEAR for both molecules and $CF^2$ in molecule 2. Both fail to adhere to valency rules, a behavior also noted in Sec. 4.4.

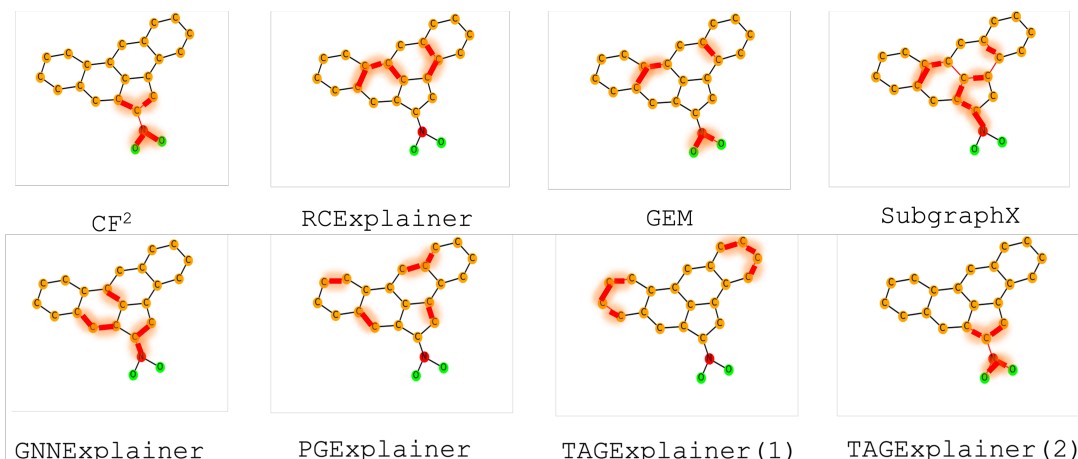

Figure N: Visualization of factual explanations on a mutagenic molecule from the Mutag dataset. The explanations contain the edges highlighted in red.

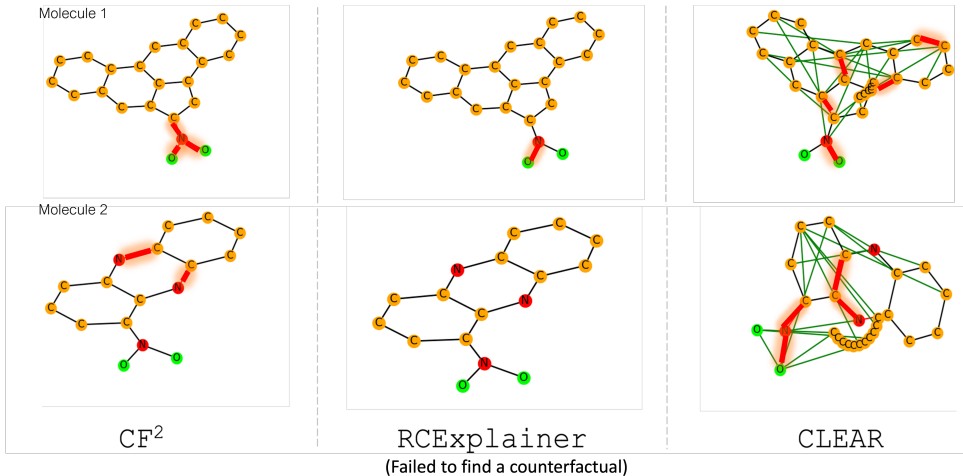

Figure O: Visualization of counterfactual explanations on Mutag dataset. Edge additions and deletions are represented by green and red colors respectively.

## G   ADDITIONAL EXPERIMENTS ON SPARSITY METRIC

**Counterfactual explainers:** Sparsity is defined as the proportion of edges from the graph that is retained in the counter-factual Yuan et al. (2022); a value close to 1 is desired. In node classification, we compute this proportion for edges in the $\mathcal{N}_v^\ell$, i.e., the $\ell$-hop neighbourhood of the target node $v$. We present the results on sparsity metric on node and graph classification tasks in Table V and W,

respectively. For node classification, we see CF-GNNEXPLAINER continues to outperform (Table V). The results are consistent with our earlier results in Table 6. Similarly, Table W shows that RCExplainer continues to outperform in the case of graph classification (earlier results show a similar trend in Table 5).

**Factual explainers:** For experiments with factual explainers, we report results of the necessity metric on varying degrees of sparsity (Recall Fig. K). Note that size acts as a proxy for sparsity in this case. This is because sparsity only involves the normalized size where it is normalized by the number of total edges. So sparsity can be obtained by normalizing all perturbation sizes in the plots to get the sparsity metric. For counterfactual explainers, we do not supply explanation size as a parameter. Hence, computing the sparsity of the predicted counterfactual becomes relevant.

Table V: Results on sparsity of counterfactual explainers for node classification. CF-GNNExplainer consistently produces the best results that are shown in gray.

| Method/Dataset | Tree-Cycles | Tree-Grid | BA-Shapes |
|---|---|---|---|
| CF-GNNEXPLAINER | 0.93 ±0.02 | 0.95 ±0.03 | 0.99 ±0.00 |
| CF$^2$ | 0.52 ±0.14 | 0.58 ±0.14 | 0.99 ±0.00 |

Table W: Results on sparsity of counterfactual explainers for graph classification. Best results are shown in gray. RCEXPLAINER consistently outperforms the other methods.

| Method/Dataset | Mutagenicity | Mutag | Proteins | AIDS | IMDB-B | ogbg-molhiv |
|---|---|---|---|---|---|---|
| RCExplainer | 0.96 ±0.00 | 0.94 ±0.01 | 0.94 ±0.02 | 0.91 ±0.00 | 0.98 ±0.00 | 0.96 +- 0.00 |
| CF$^2$ | 0.90 ±0.01 | 0.94 ±0.0 | NA | 0.99 ±0.01 | 0.89 ±0.04 | 0.62 ±0.05 |
| CLEAR | OOM | 0.88 ±0.07 | OOM | 0.66 ±0.04 | 0.87 ±0.06 | OOM |

## H    TAGEXPLAINER VARIANTS

TAGExplainer has two stages. We define TAGExplainer (1) as when we apply only the first stage and get explanations, whereas TAGExplainer (2) applies both stages. Figure P compares performance of these two variants.

## I    FACTUAL EXPLAINERS ON OGBG-MOLHIV

Figure R demonstrates a new dataset OGBG-Molhiv for three factual explainers. On this dataset, three factual methods are close to each other in terms of sufficiency performance.

## J    EXISTING BENCHMARKING STUDIES ON GNN EXPLAINABILITY

GraphFrameX Amara et al. (2022) and GraphXAI Agarwal et al. (2023) represent two notable benchmarking studies. While both investigations have contributed valuable insights into GNN explainers, certain unresolved investigative aspects persist.

- **Inclusion of counterfactual explainability:** GraphFrameX and GraphXAI have focused on factual explainers for GNNs. Prado-Romero et al. (2023) has discussed methods and challenges, but benchmarking on counterfactual explainers remains underexplored.
- **Achieving Comprehensive coverage:** Existing literature encompasses seven perturbation-based factual explainers. However, GraphFrameX and GraphXAI collectively assess only GnnExplainer Ying et al. (2019b), PGExplainer Luo et al. (2020), and SubgraphX Yuan et al. (2021).
- **Empirical investigations:** How susceptible are the explanations to topological noise, variations in GNN architectures, or optimization stochasticity? Do the counterfactual explanations provided align with the structural and functional integrity of the underlying domain? To what extent do these explainers elucidate the GNN model as opposed to the underlying data? Are there standout explainers that consistently outperform others in terms of performance? These are critical empirical inquiries that necessitate attention.

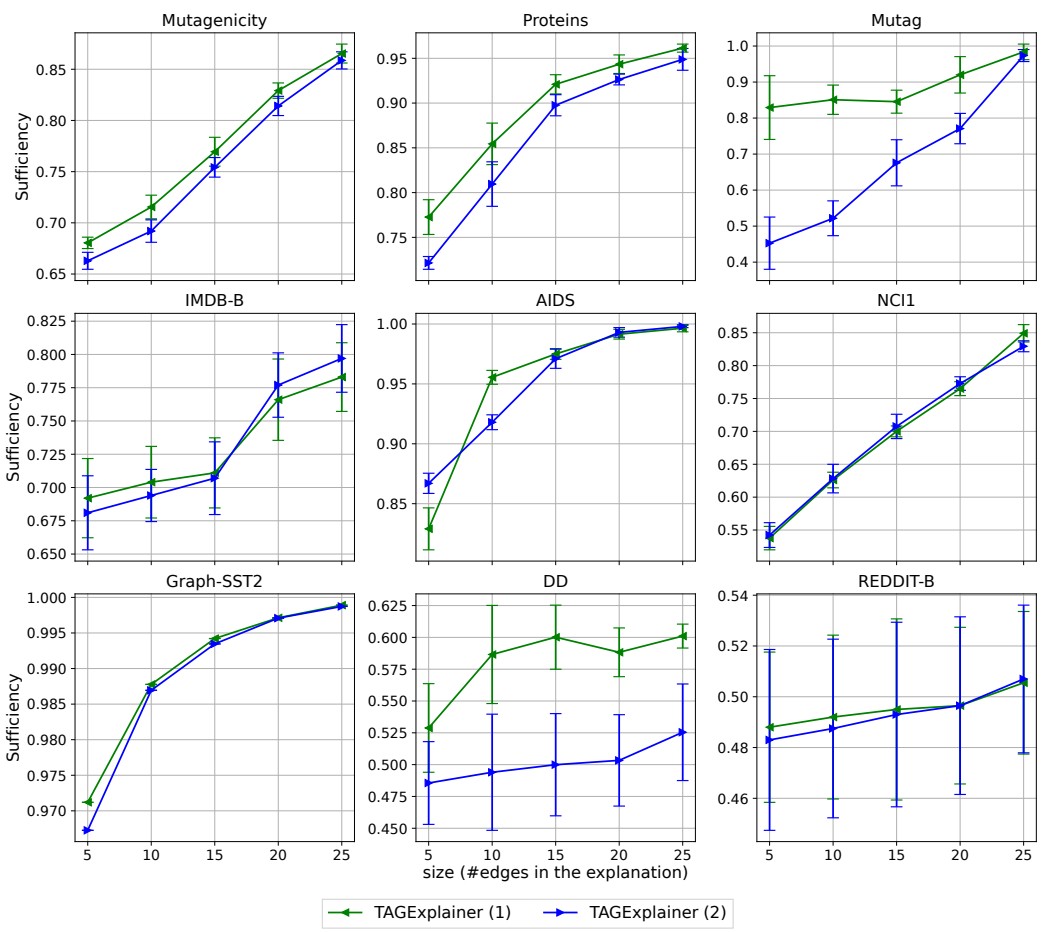

Figure P: Sufficiency of TAGExplainer variants against size. Applying the second stage does not help much for TAGExplainer.

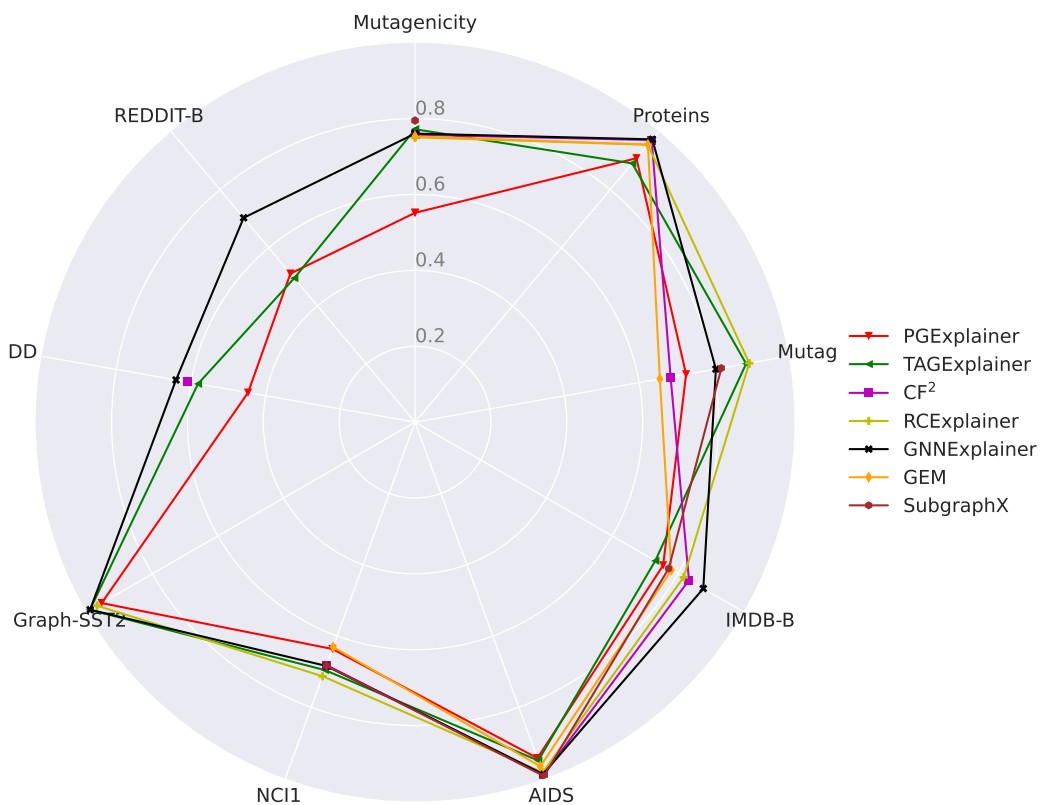

Figure Q: Spiderplot for sufficiency performance averaged for different sizes of explanations. Even though there is no clear winner method, GNNEXPLAINER and RCEXPLAINER appear among the top performers in the majority of the datasets. We omit those methods for a dataset that throw an out-of-memory (OOM) error and are not scalable.

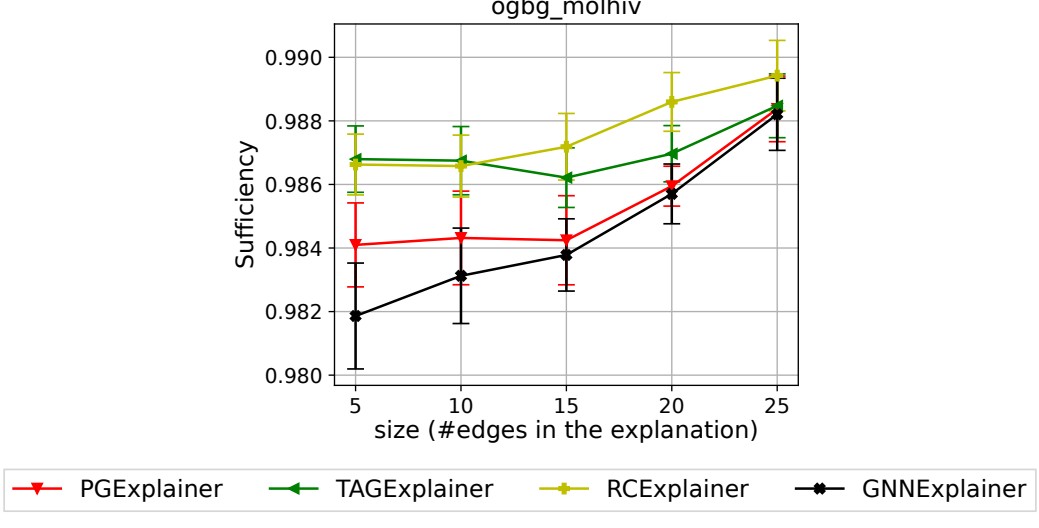

Figure R: Sufficiency of the factual explainers against the explanation size for ogbg-molhiv dataset.

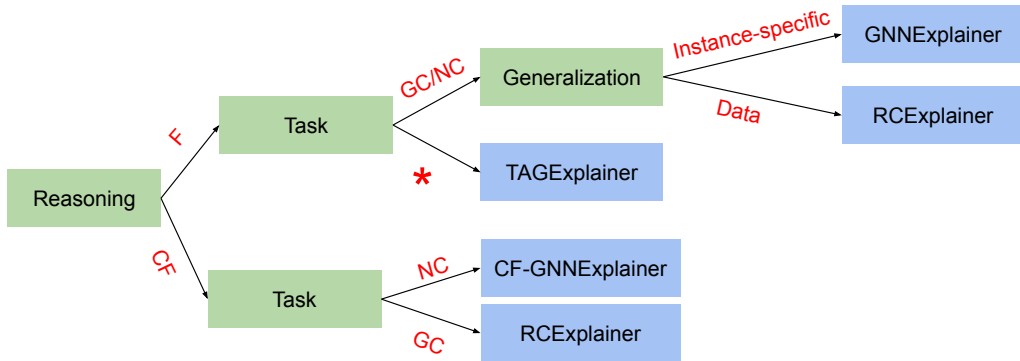

Figure S: Flowchart of our recommendations of explainers in different scenarios. Note that "F", "CF", "NC", "GC", and "*" denote factual, counterfactual, node classification, graph classification, and generalized tasks respectively.

## K    RECOMMENDATIONS IN PRACTICE

The choice of the explainer depends on various factors and we make the following recommendations.

- For counter-factual reasoning, we recommend RCExplainer for graph classification and CF-GNNExplainer for node classification.

- For factual reasoning, if the goal is to do node or graph classification, we need to first decide if we need an inductive reasoner or transductive. While an inductive reasoner is more suitable we want to generalize to large volumes of unseen graphs, transductive is suitable for explaining single instances. In case of inductive, we recommend RCExplainer, while for transductive GNNExplainer stands out as the method of choice. These two algorithms had the highest sufficiency on average. In addition, RCExplainer displayed stability in the face of noise injection.
  - For different task generalization beyond node and graph classification, one can consider using TAGExplainer.
  - In high-stakes applications, we recommend RCExplainer due to consistent results across different runs and robustness to noise.

- For both types of explainers, the transductive methods are slow. So, if the dataset is large, it is always better to use an inductive explainer over transductive ones.

While the above is a guideline, we emphasize that there is no one-size-fits-all benchmark for selecting the ideal explainer. The choice depends on the characteristics of the application in hand and/or the dataset. The above flowchart takes these factors into account to streamline the decision process. We have now added a flowchart to help the user in selecting the most appropriate. This recommendation has now also been added as a flowchart (Fig. S).

