# OpenReview forum: "GNNX-BENCH: Unravelling the Utility of Perturbation-based GNN Explainers through In-depth Benchmarking"
_ICLR.cc/2024/Conference — ICLR 2024 poster_

### Official Review · Reviewer_aTj3 · 2023-10-28

**Soundness:** 3 good
**Presentation:** 3 good
**Contribution:** 2 fair
**Rating:** 6
**Confidence:** 2

**Summary:**

This paper presents a benchmarking study on perturbation-based explainability methods for GNNs and aims to evaluate a wide range of explainability techniques.

**Strengths:**

1、This paper provides a detailed and comprehensive description of the existing explainers for GNNs and compares different explanation methods in a clear way.
2、This paper provides a large of experiments to evaluate algorithms for both factual and counterfactual reasoning explanation methods and considers the different perspectives that may affect the performance of the explanations including topology, model parameters, and model architectures.
3、The metrics for evaluating the explainers for GNNs are reasonable.

**Weaknesses:**

1、Whether adversarially adding the number of edges has an impact on the sufficiency of the factual explainers leading to different experimental results in Figure 2 or Table 4
2、Most experimental results do not seem to provide explanations about the experimental phenomena and comparisons of the advantages and disadvantages of different explanation methods. For example, why the stability of CF^2 has dropped significantly on the IMDB-B and AIDS datasets in Fig. 3.
3、For the experimental results of stability against topological noise, why are the GEM and SubgraphX not used.
4、There are differences between different factual explainers on different datasets. Is there a benchmark to select a proper explainers in practice.

**Questions:**

Please see the weaknesses stated as above.

---

> ### Author Response · Authors · 2023-11-17
> **Response to Reviewer aTj3**
>
> **Q1. Whether adversarially adding the number of edges has an impact on the sufficiency of the factual explainers leading to different experimental results in Figure 2 or Table 4.**
>
> *Response:* We have extended our stability experiments to more datasets for feature perturbation and adversarial attacks (Figures G and H, respectively). The behavior is similar to the topological noise attack explained in Section 4.2; the stability decays with the higher noise and inductive methods handle noise better than transductive.
>
> As requested, we also provide a sufficiency score with explanations generated by the topological noise attack. The results are provided in Figure F. In summary, when the noise increases, the sufficiency drops for most cases, but not in all scenarios. This shows that despite the changes in the explanations caused by the noise, GNNs may still predict the same class under noisy conditions.
>
>
>  **Q2. Most experimental results do not seem to provide explanations about the experimental phenomena and comparisons of the advantages and disadvantages of different explanation methods. For example, why the stability of CF^2 has dropped significantly on the IMDB-B and AIDS datasets in Fig. 3.**
>
> *Response:* We appreciate this feedback and now provide clear insights on trends observed across the experiments in the revised manuscript. CF^2 is a transductive method. This lies at the core of the trends observed. Specifically, transductive methods lack generalizable capability to unseen data. Furthermore, the stability is worse on denser datasets such as IMDB-B since due to the presence of more edges, the search space of explanation is larger. Transductive methods such as CF^2 and GNNExplainer witness higher deterioration in stability in such harder situations due to being transductive.
>
> **Q3. For the experimental results of stability against topological noise, why are the GEM and SubgraphX not used.**
>
> *Response:* We have updated Fig. 3 by adding GEM and SubgraphX. GEM has two-steps (distillation + generation) and one can add noise in any of these two steps. We have added noise before the distillation step, which makes it comparable to other methods. On the other hand, SubgraphX is time-consuming while providing subpar efficacy in terms of robustness compared to other explainers. For example, SubgraphX consumes $\approx 26.5$ hours to run on 435 test graphs in the Mutagenicity dataset. The inclusion of GEM and SubgraphX in Fig. 3 does not affect our conclusions.
>
> **Q4. There are differences between different factual explainers on different datasets. Is there a benchmark to select a proper explainers in practice.**
>
> *Response:* This is an excellent question and is the ultimate objective of this study! The choice of the explainer depends on various factors and we make the following recommendations. We have now added a flowchart to help the user in selecting the most appropriate. This recommendation has now also been added as a flowchart (Fig. S in App. K). The same flowchart is presented in text below. We have added this discussion in the manuscript (App. K). We have also referred this discussion in the conclusions (Sec. 5).
>
> - For counter-factual reasoning, we recommend RCExplainer for graph classification and CF-GNNExplainer for node classification.
> - For factual reasoning, if the goal is to do node or graph classification, we need to first decide if we need an inductive reasoner or transductive. While an inductive reasoner is more suitable, we want to generalize to large volumes of unseen graphs, transductive is suitable for explaining single instances. In case of inductive, we recommend RCExplainer, while for transductive GNNExplainer stands out as the method of choice. These two algorithms had the highest sufficiency on average. In addition, RCExplainer displayed stability in the face of noise injection.
> - For different task generalization beyond node and graph classification, one can consider using TAGExplainer.
> - In high-stakes applications, we recommend RCExplainer due to consistent results across different runs and robustness to noise.
> - If computation time is important, we recommend an inductive reasoner since they only involve a forward pass through the corresponding neural network. Transductive reasoners train the parameters on the input and hence is significantly slower in practice.
>
> While the above is a guideline, we emphasize that there is no one-size-fits-all benchmark for selecting the ideal explainer. The choice  depends on the characteristics of the application in hand and/or the dataset. The above flowchart takes these factors into account to streamline the decision process.

---

> > ### Author Response · Authors · 2023-11-20
> > **Looking forward to your feedback**
> >
> > Dear Reviewer,
> >
> > We thank you for taking the time to provide constructive comments, which have significantly improved the quality of the manuscript. In our revision, we have incorporated all of the given suggestions including:
> >
> > * Impact of adversarial attacks
> > * Providing a flowchart and a guideline to choose the best explainer for a specific task and dataset
> > * Elaborating on the empirical findings and adding GEM and SubgraphX to the experiments
> >
> > With these additional experiments and improved explanations, we hope we have addressed all the concerns raised by the reviewer. If there are any outstanding concerns, we request the reviewer to please raise those. Otherwise, we would really appreciate it if the reviewer could increase the score.
> >
> > Looking forward to your response.
> >
> > Thank you,
> >
> > Authors

---

> > > ### Author Response · Authors · 2023-11-21
> > > **Eagerly awaiting for feedback from Reviewer aTj3**
> > >
> > > Dear Reviewer,
> > >
> > > We thank you for the insightful comments on our work. Your suggestions have now been incorporated in our revision and we are eagerly waiting for your feedback.
> > >
> > > As the author-reviewer discussion phase is approaching its conclusion in just a few hours, we are reaching out to inquire if there are any remaining concerns or points that require clarification. Your feedback is crucial to ensure the completeness and quality of our work. Your support in this final phase, particularly if you find the revisions satisfactory, would be immensely appreciated.
> > >
> > > regards,
> > >
> > > Authors

---

> > > > ### Author Response · Authors · 2023-11-22
> > > >
> > > > Dear Reviewer aTj3,
> > > >
> > > > We sincerely appreciate your insightful comments on our manuscript. Your valuable suggestions have been incorporated into our revision, and we are eager to receive your feedback.
> > > >
> > > > With the author-reviewer discussion phase concluding shortly, we would like to check if there are any lingering concerns or areas that may require further clarification. Your feedback and evaluation play a pivotal role in determining the ultimate fate of our work. Your support in this final phase, especially if you find the revisions satisfactory, would be of immense significance.
> > > >
> > > > regards,
> > > >
> > > > Authors

---

> > > > ### Comment · Reviewer_aTj3 · 2023-11-22
> > > > **Thanks for the detailed response**
> > > >
> > > > Thanks for the detailed response and clarification of the authors. Part of my concerns have been addressed. Overall, I  will raise my score.

---

> > > > > ### Author Response · Authors · 2023-11-23
> > > > > **thank you to reviewer aTj3**
> > > > >
> > > > > Dear Reviewer aTj3,
> > > > >
> > > > > Thank you for your positive feedback on our work. If you could point out any lingering concerns, we will address those as well. We truly appreciate your engagement in the review process and helping us elevate the quality of the manuscript.
> > > > >
> > > > > regards,
> > > > >
> > > > > Sayan

---

### Official Review · Reviewer_ac1w · 2023-11-01

**Soundness:** 3 good
**Presentation:** 2 fair
**Contribution:** 4 excellent
**Rating:** 6
**Confidence:** 3

**Summary:**

This paper comprehensively studies the existing methods of explaining GNN predictors, including factual explainers and counterfactual explainers. The investigation is carried out in terms of stability, necessity/reproducibility, and feasibility for counterfactual explainers. And some conclusion of these investigations is obtained by extensive empirical evaluations.

**Strengths:**

This paper investigates a critical problem of GNN predictors, that is the explainability pertaining to GNNs. Specifically, the counterfactual explanation is an important and meaningful explainer, which is worthy of studying. This paper fulfills the vacancy of the research on benchmarking it. The empirical evaluation is extensive and comprehensive. Therefore, the resulting conclusions from the experiments are solid and reliable.

**Weaknesses:**

I believe there is some space for improvement in the paper's presentation. I suggest some important metrics, such as necessity and reproducibility can be expressed in mathematical equations. This can bring convenience for readers to understand the concepts.

**Questions:**

The notations of A(G_s) in Equation 1 should be A(G').
In definition 2, the prediction of the perturbated graph is defined to be different from the counterpart of the original graph. I think this is valid for binary classification. For multi-classification problems, the counterfactual reasoning should be associated with a pre-assigned label.
The definitions of fidelity, necessity, and reproducibility seem to be vague. The fidelity in section 4 (under figure 2) is not clear, "some works have used the term fidelity instead of sufficiency" has no citations. The authors do not clearly give the definition of necessity and reproducibility and only give the intuition of these metrics (e.g. measures if ). This makes it difficult to figure out the metrics.
The measurement of feasibility which is defined as the number of connected graphs is somehow one-sided. Are there other measurements that can characterize the topological properties? And the relationship between defining the similarity with the topological properties of the test dataset and the feasibility measurement should be discussed and justified.

---

> ### Author Response · Authors · 2023-11-17
> **Response to Reviewer ac1w: Part 1**
>
> **Q1. I believe there is some space for improvement in the paper's presentation. I suggest some important metrics, such as necessity and reproducibility can be expressed in mathematical equations. This can bring convenience for readers to understand the concepts.**
>
> Response: Thank you for this suggestion. We now define all metrics mathematically in Table 3 while discussing the metrics in Sec. 3 of the revised manuscript. The relevant metrics are reproduced verbatim below for easy access.
>
> $$
>     \text{Sufficiency}(\mathcal{S}) = \frac{\sum_{i=1}^{|\mathbb{G}|} \mathbb{1}(\Phi({\mathcal{G}_S^i}) = \Phi(\mathcal{G}^i))}{|\mathbb{G}|}
> $$
>
> $$
>     \text{Necessity}(\mathcal{N}) = \frac{\sum_{i=1}^{|\mathbb{G}|} \mathbb{1}(\Phi(\mathcal{R}^i) \neq \Phi(\mathcal{G}^i))}{|\mathbb{G}|}
> $$
>
> $$
>     \text{Stability}(\mathcal{E}_X, \mathcal{E}'_X) = \frac{\vert\mathcal{E}_X\cap\mathcal{E}'_X\vert}{\vert\mathcal{E}_X\cup\mathcal{E}'_X\vert}
> $$
>
> $$
>     \text{Reproducibility}^{+}(\mathcal{R^+}) = \frac{ACC(\Phi_S)}{ACC(\Phi)}
> $$
>
> $$
>     \text{Reproducibility}^{-}(\mathcal{R^-}) = \frac{ACC(\Phi_R)}{ACC(\Phi)}
> $$
>
> - $\mathbb{G} = \{\mathcal{G}^1, \mathcal{G}^2, \dots, \mathcal{G}^n\}$, the set of all graphs.
> - ${\mathcal{G}^i_S}$ is the explanation subgraph of $\mathcal{G}^i$
> - $\mathbb{G}_S = \{\mathcal{G}^1_S, \mathcal{G}^2_S, \dots, \mathcal{G}^n_S\}$ is the set of explanations.
> - $\mathcal{R}^i = \mathcal{G} - \mathcal{G}^i_S$
> - $\mathbb{R} = \{\mathcal{R}^1, \mathcal{R}^2, \dots, \mathcal{R}^n\}$ is the set of residual graphs.
> - $\Phi$, $\Phi_S$, and $\Phi_R$  are the models trained on $\mathbb{G}$, $\mathbb{\mathbb{G}}_S$, and $\mathbb{\mathbb{R}}$, respectively. All models are trained on the same labels.
> - $\Phi(\mathbb{G}^i)$ is the model's prediction on $\mathbb{G}^i$.
> - $\mathcal{E}_X$ and $\mathcal{E}'_X$ are the set of edges in the explanations on the original graph and after noise injection.
> - $ACC(\Phi)$ is the test accuracy of the model $\Phi$.
>
> **Q2. The notations of A(G_s) in Equation 1 should be A(G').**
>
> *Response:* We have fixed it. Thanks for pointing this out.
>
>  **Q3. In definition 2, the prediction of the perturbated graph is defined to be different from the counterpart of the original graph. I think this is valid for binary classification. For multi-classification problems, the counterfactual reasoning should be associated with a pre-assigned label.**
>
> *Response:*  Thank you for the suggestions. This has been incorporated in Def. 2. The relevant text is reproduced verbatim below.
>
> > In case of multi-class classification, if one wishes to switch to a target class label(s), then the optimization objective is modified as $\mathcal{G}^*=\arg\min_{\mathcal{G}'\in\mathbb{G},\: \Phi(\mathcal{G}')=\mathbb{C}} \; dist(\mathcal{G},\mathcal{G}')$, where $\mathbb{C}$ is the set of desired class labels and $\mathbb{G}$ is the set of all graphs one may construct by perturbing $\mathcal{G}$.
>
> **Q4. The definitions of fidelity, necessity, and reproducibility seem to be vague. The fidelity in section 4 (under figure 2) is not clear, "some works have used the term fidelity instead of sufficiency" has no citations**.
>
> *Response:* As noted in our response to Q1, we have now provided mathematical definitions of all the metrics in Table 3. Furthermore, we have removed the term Fidelity from our manuscript and consistently use only Sufficiency.

---

> > ### Author Response · Authors · 2023-11-17
> > **Response to Reviewer ac1w: Part 2**
> >
> > **Q5. The measurement of feasibility which is defined as the number of connected graphs is somehow one-sided. Are there other measurements that can characterize the topological properties? And the relationship between defining the similarity with the topological properties of the test dataset and the feasibility measurement should be discussed and justified.**
> >
> > *Response:* We conduct the analysis of feasibility only on molecular graphs. The objective is to assess if the recourse molecules produced by a counterfactual reasoner is feasible in the real world.
> >
> > In molecules, it is rare for it to be constituted of multiple connected components [1]. Hence, we study the distribution of molecules that are connected in the original dataset and compare that to the distribution in counterfactual recourses. We measure the $p$-value of this deviation. To leave no room for ambiguity, we now clearly mention that the feasibility assessment through connectivity comparison is relevant for molecular graphs only.
> >
> >  As shown in Table 10, the recourses often deviate significantly with a larger portion not forming a connected component. This result indicates chemically infeasible recourses, and therefore of less practical value. As we discuss in Sec 5 of the revised manuscript, a potential solution may lie in making feasibility an explicit component in the objective function. Modeling the feasibility score in a domain-agnostic manner can be obtained by incorporating ideas from generative modeling for graphs.
> >
> > We further note that comparing connectivity of recourses with original molecules (graphs) is just one way of studying feasibility and other mechanisms may be used (such as the number of chemical bonds added or deleted in the recourse, which we observe is massively higher for CLEAR in Table 5 indicating infeasibility). Nonetheless, it is difficult to have a single feasibility definition, since it is highly dependent on the domain. Consequently, we focus on molecules only.
> >
> > [1] Philippe Vismara and Claude Laurenço. An abstract representation for molecular graphs, pp. 343–366. 04 2000. ISBN 9780821809877. doi: 10.1090/dimacs/051/26.

---

> > > ### Author Response · Authors · 2023-11-21
> > > **keenly awaiting your feedback**
> > >
> > > Dear Reviewer,
> > >
> > > Since we are only a day away from the completion of the discussion phase, we are eagerly awaiting your feedback on the revised manuscript. We have addressed all of the presentation-related concerns identified in the review. We also made the feasibility study more comprehensive through the addition of new metrics, more explainers, and proper contextualizing in the space of molecular graphs.
> > >
> > > We would love to discuss more if any concern remains unaddressed. Otherwise, we would appreciate it if you could support the paper by increasing the score.
> > >
> > > regards,
> > >
> > > Authors

---

### Official Review · Reviewer_NT8w · 2023-11-02

**Soundness:** 3 good
**Presentation:** 3 good
**Contribution:** 3 good
**Rating:** 8
**Confidence:** 4

**Summary:**

This work presents a benchmark for perturbation-based GNN explanation methods. Within the benchmark, this work provides a comprehensive comparison between both the factual and counter-factual explanation methods. This work conducts the comparison on various datasets, including 10 graph classification and 3 node classification ones. The experiments compare 7 GNN explanation methods in terms of the size of the perturbation, the sufficiency (percentage of the subgraphs that yield the same results as using full graph), and accuracy (the percentage of correct explanation). The results give a relative comparison among the seven methods. Furthermore, this work conducts a comparison of the methods regarding the stability with regard to noise injected into the underlying graph, different seeds of training explanation models, and variations in model architectures. Lastly, this work also provides an open-sourced codebase, making the benchmark accessible for public use.

**Strengths:**

- This work provides an extensive evaluation of perturbation-based GNN explanation methods in terms of both performance and stability. The comprehensive results give a quantitative comparison of existing explanation methods.
- Besides performance, this work also considers the stabilities of GNN explanation methods. This provides a new aspect for evaluating GNN explanation algorithms.
- This work provides an online repository of the proposed benchmark, making the evaluations accessible for a broader range of users.

**Weaknesses:**

- The conclusions from the empirical comparison are not clear. It would be better to summarize the conclusions from the empirical comparison and provide conceptual insights, such as how would the results guide the future design of GNN explanation methods.
- Discussion of the existing works needs more structure. Current discussion of the related works are based on the summary for each method. It would be better to provide more structures of the current works, such as what methods the existing ones share in common.
- Details of the empirical studies need to be further elaborated. Please see Questions.

**Questions:**

- It would be better to clearly define how the sufficiency score is computed.
- Do the authors have any explanation for why the RCExplainer and the GNNExplainer have the highest sufficiency scores? It would be helpful to provide a more structured and conceptual comparison between the explanation methods.
- What is the definition and scale of the noise shown in Figure 3?

---

> ### Author Response · Authors · 2023-11-17
> **Response to Reviewer NT8w: Part 1**
>
> **Q1. The conclusions from the empirical comparison are not clear. It would be better to summarize the conclusions from the empirical comparison and provide conceptual insights, such as how would the results guide the future design of GNN explanation methods.**
>
> *Response:* This is an excellent suggestion. We appreciate the constructive feedback. To incorporate this suggestion, we have now made the following changes.
>
> * The conclusion section (Section 5) has been expanded to discuss research directions that may potentially yield solutions to the primary limitations identified in this study, namely - incorporating *generative modeling* in counterfactual reasoning to address feasibility concerns, and migrating towards *ante-hoc* reasoning to address instability and non-reproducibility. Current explainers are post-hoc in nature, wherein the explanations are generated post the completion of GNN training. In this pipeline, the explainers have no visibility to how the GNN reacts to perturbations on the input data, initialization seeds, etc. In the *ante-hoc* paradigm, the GNN and the explainer are jointly trained.
> *  For each experiment, we have added a paragraph titled "Insights" to clearly discuss the take-aways and provide explanations for the trends observed.
>
> **Q2. Discussion of the existing works needs more structure. Current discussion of the related works is based on the summary for each method. It would be better to provide more structures of the current works, such as what methods the existing ones share in common.**
>
> *Response:* We have incorporated this suggestion through the following mechanisms:
> * Tables 1 and 2 characterize different factual and counterfactual methods, respectively, along various dimensions such as scoring function, intended task (such as graph/node classification), modeling paradigm (transductive vs. inductive), etc. Through these tables, it is now easy to derive the common principles shared across algorithms.
> * We have also re-structured the related work discussion (Sec. 2.1) based on these tables to better elucidate the common design principles and enable easier abstraction of their conceptual designs.
>
> **Q3. It would be better to clearly define how the sufficiency score is computed.**
>
> *Response:* We have added the mathematical formulation of sufficiency as well as all other metrics used in our work (See Table 3 in the revised manuscript). The relevant portion is reproduced verbatim below.
>
> Sufficiency encodes the ratio of graphs for which the prediction derived from the explanation matches the prediction obtained from the complete graph.
>
> $$
>     \text{Sufficiency}(\mathcal{S}) = \frac{\sum_{i=1}^{|\mathbb{G}|} \mathbb{1}(\Phi({\mathcal{G}_S^i}) = \Phi(\mathcal{G}^i))}{|\mathbb{G}|}
> $$
>
> $$
>     \text{Necessity}(\mathcal{N}) = \frac{\sum_{i=1}^{|\mathbb{G}|} \mathbb{1}(\Phi(\mathcal{R}^i) \neq \Phi(\mathcal{G}^i))}{|\mathbb{G}|}
> $$
>
> $$
>     \text{Stability}(\mathcal{E}_X, \mathcal{E}'_X) = \frac{\vert\mathcal{E}_X\cap\mathcal{E}'_X\vert}{\vert\mathcal{E}_X\cup\mathcal{E}'_X\vert}
> $$
>
> $$
>     \text{Reproducibility}^{+}(\mathcal{R^+}) = \frac{ACC(\Phi_S)}{ACC(\Phi)}
> $$
>
> $$
>     \text{Reproducibility}^{-}(\mathcal{R^-}) = \frac{ACC(\Phi_R)}{ACC(\Phi)}
> $$
>
> - $\mathbb{G} = \{\mathcal{G}^1, \mathcal{G}^2, \dots, \mathcal{G}^n\}$, the set of all graphs.
> - ${\mathcal{G}^i_S}$ is the explanation subgraph of $\mathcal{G}^i$
> - $\mathbb{G}_S = \{\mathcal{G}^1_S, \mathcal{G}^2_S, \dots, \mathcal{G}^n_S\}$ is the set of explanations.
> - $\mathcal{R}^i = \mathcal{G} - \mathcal{G}^i_S$
> - $\mathbb{R} = \{\mathcal{R}^1, \mathcal{R}^2, \dots, \mathcal{R}^n\}$ is the set of residual graphs.
> - $\Phi$, $\Phi_S$, and $\Phi_R$ are the models trained on $\mathbb{G}$, $\mathbb{\mathbb{G}}_S$, and $\mathbb{\mathbb{R}}$, respectively. All models are trained on the same labels.
> - $\Phi(\mathbb{G}^i)$ is the model's prediction on $\mathbb{G}^i$.
> - $\mathcal{E}_X$ and $\mathcal{E}'_X$ are the set of edges in the explanations on the original graph and after noise injection.
> - $ACC(\Phi)$ is the test accuracy of the model $\Phi$.

---

> > ### Author Response · Authors · 2023-11-17
> > **Response to Reviewer NT8w: part 2**
> >
> > **Q4. Do the authors have any explanation for why the RCExplainer and the GNNExplainer have the highest sufficiency scores? It would be helpful to provide a more structured and conceptual comparison between the explanation methods.**
> >
> > *Response:* GNNExplainer is transductive, wherein it trains the parameters on the input graph itself. In contrast, inductive methods use pre-trained weights to explain the input. Consequently, transductive methods, such as GNNExplainer, at the expense of higher computation cost, has an inherent advantage in terms of optimizing sufficiency. Compared to other transductive methods, GNNExplainer utilizes a loss function that aims to increase sufficiency directly. This makes the method a better candidate for sufficiency compared to other inductive and transductive explainers. On the other hand, for RCExplainer, we believe that calculation of decision regions for classes helps to increase its generalizability as well as robustness. Finally, as a trend, both methods provide consistent sufficiency increase with the size of explanations for all datasets, while we observe an oscillation in sufficiency in Mutag for other explainers. This makes RCExplainer and GNNExplainer more reliable as well.
> >
> > We have incorporated the above discussion while analyzing the sufficiency results in Fig. 2.
> >
> >  **Q5. What is the definition and scale of the noise shown in Figure 3?**
> >
> > *Response:* In Figure 3, the xticks (Noise) denote the number of perturbations made to the edge set of the original graph. Here, perturbations include randomly sampling $x$ (denoted on x-axis) negative edges and adding them to the original edge set (i.e., connect a pair of nodes that were previously unconnected). We now mention this explicitly in the caption of Fig. 3.
> >
> > We note that we also explore adding noise through adversarial attack, where edges are both removed and added. These experiments are in Appendix C.

---

> > > ### Author Response · Authors · 2023-11-21
> > > **Keenly awaiting your feedback**
> > >
> > > Dear Reviewer,
> > >
> > > Since we are only a day away from the completion of the discussion phase, we are eagerly awaiting your feedback on the revised manuscript. We have addressed all of the presentation related concerns and present a significantly expanded discussion on potential solutions to the previously undiscovered shortcomings of the explainability methods with regards to stability, feasibility and reproducibility.
> > >
> > > Please do let us know if there are any additional clarifications or experiments that we can offer. We would love to discuss more if any concern still remains. Otherwise, we would appreciate it if you could support the paper by increasing the score.
> > >
> > > regards,
> > >
> > > Authors

---

> > > > ### Comment · Reviewer_NT8w · 2023-11-22
> > > > **Thanks for the authors' responses**
> > > >
> > > > I have read the authors' responses. The authors have added more structures to the discussion of related works and experimental results. Thus, I would like to increase my score. After reading the experimental insights, I think that it would be better to shorten the insights and summarize them as a few sentences at the beginning of the experiments.

---

> > > > > ### Author Response · Authors · 2023-11-22
> > > > > **thank you for your positive feedback**
> > > > >
> > > > > Dear Reviewer NT8w,
> > > > >
> > > > > We thank you for your positive feedback and helping us improve the manuscript. We will definitely incorporate your suggestion on restructuring the insights.
> > > > >
> > > > > regards,
> > > > >
> > > > > Authors

---

### Official Review · Reviewer_PPUE · 2023-11-08

**Soundness:** 3 good
**Presentation:** 3 good
**Contribution:** 2 fair
**Rating:** 3
**Confidence:** 4

**Summary:**

The authors applied a benchmark evaluation to measure various GNN explainers' performances. A brief review of GNN explainer classification is introduced, then seven perturbation-based factual explainers and four perturbation-based counterfactual explainers are selected to conduct the benchmark test. Stability, necessity, reproducibility, feasibility, and comparative analysis are chosen to be evaluation criteria.  Size, fidelity, and accuracy are regarded as metrics. Finally, after the empirical evaluation, the authors provided some insights and directions expected to lead researchers enhance the overall quality and interpretability of GNNs.

**Strengths:**

1) A comprehensive introduction about how GNN explainers work and how GNN explainers are classified is provided.  Clear Figure 1 demonstrates research interest of the paper.
2) Various perturbation-based GNN explainers are selected to evaluate their performances, increasing the soundness of conclusions. The selected explainers are published from 2003 to 2022, covering the development of GNN explainers for decades.Multiple runs were conducted to deal with randomness.
3) Detailed appendix proves the rigor of experiments.  Comparative analysis is conducted to reveal some kay outcomes' features.  Many clear figures demonstrate the reasonability of conclusion.

**Weaknesses:**

1) Lack of creativity and significance: The authors conducted an evaluation to measure many explainers' performances, but did not come up with a novel solution to overcome the discovered challenges.  Also, the conclusions are not insightful enough with further analysis. Based on this reason, it is hard to regard this paper as a research paper.
2) Poor layout: The layout of words and tables is too dense, such as Table 1, 4, 5, and 8. Numbers inside table are too small and hard to read.

**Questions:**

Authors may want to try to offer some new solutions to the discovered limitations of existing GNN explainers. Finding out these limitations is just a start of a research work. For instance, the feasibility concerns about counterfactual explainations appear due to the deviations in topological distribution. Authors can keep working in this direction and find out some methods to improve feasibility.

---

> ### Author Response · Authors · 2023-11-17
> **Response to Reviewer PPUE**
>
> **Q1. Lack of creativity and significance: The authors conducted an evaluation to measure many explainers' performances, but did not come up with a novel solution to overcome the discovered challenges. Also, the conclusions are not insightful enough with further analysis. Based on this reason, it is hard to regard this paper as a research paper.**
>
> *Response:* The Call for Papers for ICLR 2024 includes 'Datasets and Benchmarks' (refer to https://iclr.cc/Conferences/2024/CallForPapers), and our submission falls within this category. Benchmarking papers are not anticipated to present novel solutions; rather, the focus is on conducting a comprehensive empirical evaluation in a relevant area and offering fresh insights previously undiscovered. It is a reasonable expectation from a benchmarking paper to propose potential directions for addressing identified limitations, as we have done in Section 5. Based on the suggestions provided, we have further expanded our discussion on potential solutions in the revised manuscript.
>
>
> **Q2. Poor layout: The layout of words and tables is too dense, such as Table 1, 4, 5, and 8. Numbers inside table are too small and hard to read.**
>
> *Response:* We appreciate this constructive feedback. We have thoroughly revisited all sections of the paper and have ensured the fonts are large enough to be comfortably legible in the revised manuscript. We have also increased spacing around tables. We hope these changes have effectively addressed the presentation-related concerns.

---

> ### Author Response · Authors · 2023-11-21
> **Looking forward to feedback from Reviewer PPUE**
>
> Dear Reviewer,
>
> Since we are only a day away from the completion of the discussion phase, we are eagerly awaiting your feedback on the revised manuscript. We have addressed all of the presentation related concerns and present a significantly expanded discussion on potential solutions to the previously undiscovered shortcomings of the explainability methods with regards to stability, feasibility and reproducibility.
>
> Finally, we wish to clarify again that our submission has been made to the Dataset and Benchmark category of ICLR.
>
> Please do let us know if there are any additional clarifications or experiments that we can offer. We would love to discuss more if any concern still remains. Otherwise, we would appreciate it if you could support the paper by increasing the score.
>
> regards,
>
> Authors

---

> > ### Author Response · Authors · 2023-11-22
> > **Eagerly waiting for feedback from Reviewer PPUE**
> >
> > Dear Reviewer PPUE,
> >
> > As the author-reviewer discussion phase is nearing its conclusion, we would like to inquire if there are any remaining concerns or areas that may need further clarification. Your support during this final phase, especially if you find the revisions satisfactory, would be of great significance. Your feedback and evaluation hold a pivotal role in determining the ultimate fate of our work. Additionally, we are pleased to inform you that one reviewer has positively responded to our revision and has increased the score to "8: Accept, Good paper."
> >
> > Regards,
> >
> > Authors

---

> > > ### Author Response · Authors · 2023-11-23
> > > **Earnestly waiting for feedback from Reviewer PPUE**
> > >
> > > Dear Reviewer PPUE,
> > >
> > > We earnestly appeal to you to share your feedback on our revised version. We are happy to inform you that all of the three reviewers now lean towards acceptance, including one noting "8: accept, good paper". Your insights and evaluation play a crucial role in deciding the ultimate fate of our work, and we are eagerly awaiting your response to the revised manuscript.
> > >
> > > regards,
> > >
> > > Authors.

---

> > > > ### Author Response · Authors · 2023-11-23
> > > > **Final appeal to Reviewer PPUE**
> > > >
> > > > Dear Reviewer PPUE,
> > > >
> > > > The author-reviewer discussion phase closes in 4 hours from now. Your final opinion on our revision is crucial for the decision on our paper.
> > > >
> > > > We are making one last sincere appeal for your acknowledgement of our revision and feedback on any outstanding concerns.
> > > >
> > > > Regards,
> > > >
> > > > Authors

---

### Author Response · Authors · 2023-11-17
**Overview of rebuttal**

We thank the reviewers for their insights and constructive suggestions. A comprehensive point-by-point response to the reviewers' comments is presented below. We have updated the main manuscript and the appendix to address these comments. The changes made in the manuscript are highlighted in *blue* color. The *major additional changes* are listed below.

* **Additional experiments:** We have incorporated all of the additional experiments requested by the reviewers. They include stability against adversarial attacks, and a more comprehensive assessment of feasibility of counterfactual recourses.
* **Enhancements in Presentation:** We have undertaken a series of presentation enhancements, which includes the following aspects:
    1. A more rigorous mathematical formulation of the metrics has been included for clarity.
    2. More detailed discussion on the shortcomings of existing explainers and potential avenues to overcome them. In addition, we have expanded the analysis of the various experiments conducted with particular focus on the causes behind the trends observed. Finally, we also include a flowchart that could guide users in selecting the most appropriate explainer for their dataset and task.
    3. A re-structured related work discussion to better surface the common principles shared across algorithms.

We hope these revisions will satisfactorily address the concerns raised by the reviewers and elevate the overall quality of our work.

---

### Author Response · Authors · 2023-11-20
**Keenly awaiting feedback on the rebuttal**

Dear Reviewers,

Thank you once again for all of your constructive comments, which have helped us significantly improve the paper! As detailed below, we have performed several additional experiments and analyses to address the comments and concerns raised by the reviewers.

Since we are into the last two days of the discussion phase, we are eagerly looking forward to your post-rebuttal responses.

Please do let us know if there are any additional clarifications or experiments that we can offer. We would love to discuss more if any concern still remains. Otherwise, we would appreciate it if you could support the paper by increasing the score.

Thank you!

Authors

---

### Author Response · Authors · 2023-11-22
**Eagerly awaiting for feedback**

Dear Reviewers,

As we approach the conclusion of the author-discussion phase, we express our gratitude for the valuable comments you have provided on our work. We have diligently considered and integrated your insights into our revised manuscript. Your feedback at this juncture holds immense significance for the evaluation of our research. We kindly urge you to bring forth any remaining concerns. Alternatively, your support in this final phase, especially if you find the revisions satisfactory, would be greatly appreciated.
regards,

Authors

---

### Meta-Review · Area_Chair_mynX · 2023-12-12

**Metareview:**

The authors provide a comprehensive empirical evaluation of explainability tools for GNNs. Despite some original concerns by reviewers, most reviewers remain positive about the paper. Given that this paper is submitted to the datasets and benchmarks area, the one remaining concern around lack of novelty of proposed methods is less of a concern. Therefore, the paper should be incorporated in the conference is possible.

**Justification For Why Not Higher Score:**

Reviewers found the evaluation reasonable but not exceptional

**Justification For Why Not Lower Score:**

The paper contributes meaningfully to the datasets and benchmarks area.

---

### Decision · Program_Chairs · 2024-01-16

Accept (poster)